# Protonation-State Dependence of Hydration and Interactions in the Two Proton-Conducting Channels of Cytochrome c Oxidase

**DOI:** 10.3390/ijms241310464

**Published:** 2023-06-21

**Authors:** Rene F. Gorriz, Senta Volkenandt, Petra Imhof

**Affiliations:** 1Department of Physics, Freie Universität Berlin, Arnimallee 14, 14195 Berlin, Germany; rene.gorriz@fau.de; 2Computer Chemistry Center, Friedrich-Alexander Universität (FAU) Erlangen-Nürnberg, Nägelsbachstrasse 25, 91052 Erlangen, Germany; senta.volkenandt@fau.de

**Keywords:** Cytochrome c Oxidase, protonation and hydration, molecular dynamics simulations

## Abstract

Cytochrome c Oxidase (CcO), a membrane protein of the respiratory chain, pumps protons against an electrochemical gradient by using the energy of oxygen reduction to water. The (“chemical”) protons required for this reaction and those pumped are taken up via two distinct channels, named D-channel and K-channel, in a step-wise and highly regulated fashion. In the reductive phase of the catalytic cycle, both channels transport protons so that the pumped proton passes the D-channel before the “chemical” proton has crossed the K-channel. By performing molecular dynamics simulations of CcO in the O→E redox state (after the arrival of the first reducing electron) with various combinations of protonation states of the D- and K-channels, we analysed the effect of protonation on the two channels. In agreement with previous work, the amount of water observed in the D-channel was significantly higher when the terminal residue E286 was not (yet) protonated than when the proton arrived at this end of the D-channel and E286 was neutral. Since a sufficient number of water molecules in the channel is necessary for proton transport, this can be understood as E286 facilitating its own protonation. K-channel hydration shows an even higher dependence on the location of the excess proton in the K-channel. Also in agreement with previous work, the K-channel exhibits a very low hydration level that likely hinders proton transfer when the excess proton is located in the lower part of the K-channel, that is, on the N-side of S365. Once the proton has passed S365 (towards the reaction site, the bi-nuclear centre (BNC)), the amount of water in the K-channel provides hydrogen-bond connectivity that renders proton transfer up to Y288 at the BNC feasible. No significant direct effect of the protonation state of one channel on the hydration level, hydrogen-bond connectivity, or interactions between protein residues in the other channel could be observed, rendering proton conductivity in the two channels independent of each other. Regulation of the order of proton uptake and proton passage in the two channels such that the “chemical” proton leaves its channel last must, therefore, be achieved by other means of communication, such as the location of the reducing electron.

## 1. Introduction

Cytochrome c Oxidase (CcO) is a membrane protein and part of the respiratory chain (also known as complex IV). Receiving electrons from cytochrome c, it undergoes a redox cycle in which it transforms dioxygen and protons into water. In addition to four protons, thus consumed, CcO pumps four additional protons from the inside of the cell to the outside of the membrane and against a proton gradient [1]. The resulting increase in electrochemical membrane potential can then be used by ATPases to synthesise ATP.

Uptake of the protons and transport to the bi-nuclear redox centre (BNC) for water production (“chemical” protons), and uptake of those protons pumped can occur via two distinct pathways, the D- and K-channels, named after important residues, D132 and K362 (*Rh. sphaeroides* numbering), respectively. The D-channel connects the N-side of the membrane from D132 to E286. The K-channel spans from E101 or H96 on the N-side to Y288 at the BNC.

Both “chemical” and pumped protons pass only through the D-channel in the oxidative phase (states A to F in Figure 1) [2,3,4]. In the reductive phase of the catalytic cycle (states O to R; see Figure 1), the K-channel transports “chemical” protons, while the pumped protons pass through the D-channel [5].

For CcO to function as both an oxidase enzyme and a proton pump, the transport of protons through one or the other channel must be highly regulated. We have previously shown that in the $P_{R}$ state (between $P_{M}$ and F of the oxidative phase, and the first state in which an electron has arrived at the BNC; see Figure 1), proton passage through the D-channel is auto-regulated, preventing the transfer of an unsolicited second proton before the first one has left the D-channel towards the P-side [7]. This regulation is mainly achieved by a protonated E286 at the end of the D-channel, leading to directionality of the hydrogen bonds in a water network that favours the closed conformation of the so-called asparagine gate (formed by residues N121 and N139), thereby preventing the formation of a hydrogen-bonded chain of water molecules that allows proton passage to the upper part of the D-channel. The asparagine gate and, with that, proton transport in the D-channel are further affected by the K-channel such that the protonation of K362, located about half-way up the K-channel, favours longer-lived hydrogen bonds between N139 and water molecules and, with that, the formation of a hydrogen-bonded water wire, only with the excess proton located in the lower half of the D-channel.

In a structure of oxidised CcO, there is no connection between the end of the D-channel and the BNC by water molecules [8], confirming that the transport of “chemical” protons to the BNC in the reductive phase has to go through the K-channel. The proton passage through the K-channel in the O→E redox state, after the arrival of the first electron in the reductive phase, depends on whether the channel contains sufficient water molecules such that a hydrogen-bonded network of protein residues and water molecules can span all the way from the channel entrance to the BNC. The hydration level, in turn, depends on the location of the excess proton. Our previous simulation study of CcO in the O→E redox state showed the transfer of a proton past S365 (at about the lower third of the K-channel) to be determinant for an increase in channel hydration that allows proton transport further up the K-channel [6].

There is an ongoing controversy in the literature about the order of events, that is, electron transfer and transport of the protons, in the O→E transition [9]. Based on kinetic absorption spectra and electrometric techniques [10], it was proposed that the pumped proton passes the D-channel before the reducing electron reaches the BNC. This idea was challenged, since this proton transfer to the proton loading site could take place without a reducing electron being transferred to the BNC [11]. Therefore, Wikström and co-workers proposed electron transfer to the BNC prior to the arrival of the proton at the proton loading site, enhanced by a positive charge in the K-channel close to the BNC [11]. This charge has further been proposed to be located on K362 in a conformation pointing towards the BNC (“up” conformation). Time-resolved optical spectroscopy and EPR experiments on a K362M mutant showed electron transfer to the BNC but not fully from heme a3 to CuB [12]. There is a consensus, though, that proton delivery through the K-channel can only be achieved with K362 present and a proton arriving at the BNC expels the (then pumped) proton from the proton loading site.

In studies on the pH gradient across the membrane, it was shown that the states of protonation of key residues in the two channels are not independent of each other [13]. In particular, when Y288 has low proton affinity, the protonation states of E286 and K362 are anti-correlated, which can be understood as only one channel being active at a time. Upon the protonation of Y288, this anti-correlation was found to be lost, suggesting that both channels could operate simultaneously. However, a protonated Y288 renders the simultaneous protonation of K362 unlikely [13] and corresponds to a completed passage of a proton through the K-channel. Since the simultaneous protonation of Y288 and E286 has also been found to be unlikely [13], the D-channel becomes inactive upon the protonation of Y288. The passage of a proton through the D-channel can, therefore, only take place before the arrival of the “chemical” proton at Y288.

The transport of protons through one or the other channel must thus take place in the correct order. Should the “chemical” proton pass through the K-channel, reach the BNC, and be consumed in the redox reaction to water before a proton has reached the proton loading site, the enzyme would have missed the possibility to pump a proton across the membrane. It is, however, unclear how the two channels are orchestrated to achieve the correct order of proton transfers.

In this work, we used molecular dynamics simulations to simulate CcO in different protonation states of the D- and K-channels and analysed the communication within and between the protein residues of the two channels ultimately affecting the proton transfer probability in one or the other channel.

## 2. Results

### 2.1. Conformation Analysis

The most remarkable differences in the distances between D-channel residues in the different models can be observed for the distances between N139 and E286, which are affected by the protonation state of E286 (see Figure 2). That is, in models with protonated E286 (models 01* and 11*), the distances between the OD1 atoms of N139 and E286 are predominantly shorter than those in models with unprotonated E286 (models 00* and 10*), corresponding to a conformation of N139 with the OD1 atom pointing towards E286 in the former case and away in the latter case (see also Figure 3).

Accordingly, the two asparagine residues, N121 and N139, are predominantly closer to each other in models with protonated E286 (see Figure 4). In some models, sub-populations with lower probability of larger/shorter distances between N139 and E286, and between N121 and N139, respectively, are also observed, but the major conformations clearly depend on the protonation state of E286. (Distributions of side chain dihedral angles of residues in the d-channel are reported in Appendix A.) The protonation of D132, in contrast, hardly affects the distances between D-channel residues, including distances from D132 itself (see Appendix A).

In the K-channel (for snapshots, see Figure 5), variations in the distances between protein residues are observed upon the protonation of K362 (models *10 and *11), which then exhibit shorter distances from Y288 and T359, corresponding to an “up” conformation of K362 (see Appendix A and Figure 6 and Appendix A). E101 also shows shorter distances from Y288 (see Appendix A) and, accordingly, an “up” conformation when protonated (models *1 ) and is thus not repelled by the negative charge of Y288. This ss also the case in models in which a $H_{3}O^{+}$ ion is close to Y288 (models *00f). With a $H_{3}O^{+}$ ion close to E101 (models *00b) and in models which also have K362 protonated (models *11), E101 exhibits both “up” and “down” conformations, which are reflected in the distances from Y288 and from K362 (see Figure 7 and Figure 8, respectively. For further distances between K-channel residues see Appendix A). The dependence of the K362 and E101 conformations on the protonation state of the K-channel is also apparent from the distribution of the side-chain dihedral angles of the two residues (see Appendix A, respectively, and Appendix A for dihedral angles of T359, S365 and H96).

Regarding the distances between residues of the two channels, K362 and N139 differ in their distances, depending on the protonation state of the respective channels. That is, while the conformation of K362, and thus its distance from N139, is only affected by its own protonation, the conformation of N139 depends on the protonation of E286. The results are slightly shorter distances between the two residues when E286 is unprotonated (models 00* and 10*) and K362 is in a “down” conformation, which is the case in models with unprotonated K362 and no $H_{3}O^{+}$ ion in the K-channel (models *01 and *11) or a $H_{3}O^{+}$ ion in the lower part of the K-channel (models *00a, *00b, and *00c). Correspondingly, the K362-N139 distances are larger when K362 is in an “up” conformation, i.e., in models in which K362 is protonated (models *10 and *11) or which has an $H_{3}O^{+}$ ion in the upper part of the K-channel (models *00e and *00f) (see Figure 5, Figure 9, and Appendix A). It is interesting to note that the two residues exhibit the shortest distances in models with a
$H_{3}O^{+}$ ion close to E101 (models *b), where K362 is clearly in a “down” conformation, as can also been seen from its distances from Y288 (see Figure 6). Further distance distributions are reported in Appendix A.

### 2.2. Electrostatic Interactions

The electrostatic interactions between residues in the D- and K-channels of CcO are, as anticipated, clearly affected by the titratable residues being charged or not. That is, unprotonated D132 and unprotonated E286 (models 00*) exhibit unfavourable electrostatic interactions despite their large distance (∼29 Å; cf. Appendix A). In contrast, when E286 is protonated or both residues are neutral (models 01* and 11*, respectively), their electrostatic interactions are negligible (see Figure 10). It is interesting to note that the interactions between E286 and non-titratable N139 are also non-negligible only with charged E286 (models 00* and 10*), albeit with large errors (see Figure 10).

The electrostatic interactions between K362 and Y288 are the strongest among all pairs analysed when K362 is protonated (see Figure 10). In those cases, not only two oppositely charged residues interact, but also the distance between the two is shorter than that in models with neutral K362 (∼10 Å and ∼11–13 Å, respectively; see Appendix A). Shorter K362-Y288 distances (∼8 Å and ∼10 Å, respectively; see Appendix A) can also be observed in models in which the
$H_{3}O^{+}$ ion is close to Y288 (models *00f) or just above K362 (models *00f). In these models, however, the K362-Y288 interaction is even weaker than that in the other models with neutral K362 (models *00 and *01), likely due to a screening effect by the
$H_{3}O^{+}$ ion. Similarly, K362 interacts most favourably with E101 when both residues sre (oppositely) charged (models *10 and *01; see Figure 10).

E286 and K362, with a distance of ∼18–19 Å (see Appendix A) and located in different channels, interact strongly and favourably if both residues are (oppositely) charged (E286 unprotonated and K362 protonated (models 0010, 0011, 1010, and 1011)) and show very small favourable electrostatic interactions if only K362 is charged (models 0110, 0111, 1110, and 1111; see Figure 10). In addition, the pair K362-D132 interacts most favourably when both residues are oppositely charged (models 0010, 0011, 0110, and 0111) but much more weakly due to a considerably larger distance between the residues (∼28–31 Å; see Appendix A). E286 and Y288, which are separated by ∼13 Å (see Appendix A), show highly unfavourable electrostatic interactions for unprotonated E286 (see Figure 10) due to both residues being negatively charged.

In some models, there is also a weak electrostatic interaction between K362 and N139. This is favourable in models 0110, 0111, 1110, and 1111, which all have protonated K362 and protonated E286. This can be explained by N139 being preferable in conformations in which the the OD atom points towards E286 (see Figure 3), hence with the more negatively charged side-chain atom being closer to the (positively charged) K362 when E286 is protonated. The opposite effect is less pronounced, but models 1010 and 1011 with unprotonated E286 and protonated K362 show weak unfavourable interactions.

The $H_{3}O^{+}$ ion itself interacts very strongly and favourably with E101 (∼100 kcal/mol) when the $H_{3}O^{+}$ ion is located in the lower part of the K-channel, hence close to E101 (models *00a and *00b; see Figure 11). Once located above S365 (models *00c), these interactions drop significantly but are still very favourable (∼−30 kcal/mol). The location of the $H_{3}O^{+}$ ion higher up in the K-channel further reduces the electrostatic interactions with E101, in agreement with their distance becoming larger (see Appendix A). Simultaneously, the electrostatic interactions of the $H_{3}O^{+}$ ion and Y288 increase as the distances from this residue become shorter, with the $H_{3}O^{+}$ ion being located higher up in the K-channel (see Figure 11). The $H_{3}O^{+}$ ion also interacts very favourably with K362 when it is very close to it in models *00d and *00e (see Figure 11).

The $H_{3}O^{+}$ ion also exhibits favourable interactions with charged E286 in the D-channel, with the strength strictly depending on the location of the $H_{3}O^{+}$ ion and increasing in models *00a to *00f. Of note, in models with protonated E286, there is also a weak favourable interaction of the $H_{3}O^{+}$ ion with N139, which increases with the $H_{3}O^{+}$ ion approaching the top of the K-channel. This can be explained by the N139 “up” conformations of the OD atom in those models, as already noted for interactions of N139 and K362. Similarly, increasing but weak repulsive interactions between N139 and the $H_{3}O^{+}$ ion are observed when E286 is unprotonated and N139 exhibits a different conformation.

### 2.3. Channel Hydration

As can be seen from Table 1 and Appendix A, the number of water molecules in the D-channel is clearly affected by the protonation state of E286, with about four to five water molecules more in models with charged E286 (compare models 00* and 01*, and models 10* and 11*). In contrast, the protonation of D132 hardly has an effect on the number of water molecules observed in the D-channel (see Table 1 and Appendix A). It should be noted that by construction, the cylinder contains one to two more water molecules than the polyhedron (see Table 1 and Appendix A). The volume of the D-channel polyhedron does not vary significantly with the protonation state of the D-channel residues (see Appendix A) suggesting that it can also readily accommodate a larger number of water molecules.

Further analysis of the average positions of the water molecules, as shown in the height profiles (Figure 12) and the two-dimensional projections (Figure 13 and Appendix A), reveals that the additional water molecules in the models with charged E286 (models 00* and 10*) are mainly located in the upper part of the D-channel, that is, close to E286. At the height of and just below the so-called asparagine gate, formed by residues N121 and N139, there are significantly fewer water molecules than in the remainder of the D-channel, irrespective of the protonation model. This is mainly due to the volume excluded by the two asparagine residues. Water molecules are observed, however, at the height of N121 and at the height between the two residues N121 and N139, indicating that water can pass through or around the asparagine gate. Variation in the protonation state of the K-channel shows no significant effect on the D-channel volume nor on its hydration level.

Similarly, the hydration of the K-channel is mainly affected by its own protonation state and the position of the $H_{3}O^{+}$ ion, if present (see Table 2 and Appendix A). As also previously observed [6], the protonation of K362 is associated with a high hydration level, but only when E101 is not simultaneously protonated (models *01 but not models *11). The protonation of E101 alone, or a K-channel without any excess proton (models *01 or *00), and models with a $H_{3}O^{+}$ ion located in the lower part of the K-channel (*00a and *00b) show the lowest hydration level of the K-channel, with six–seven water molecules (Table 2), which is the number of water molecules observed in the crystal structure [14]. These water molecules are located in the lower and top-most part of the K-channel, and the probability to find a water molecule between S365 and K362 is almost negligible (see Figure 14, Figure 15 and Appendix A). The protonation of the upper part of the K-channel (models *00e and *00f, with the $H_{3}O^{+}$ ion being located close to T359 and Y288, respectively) also show rather low hydration, with about seven–nine water molecules in the K-channel, which are not enough to connect the upper and lower part of the K-channel (see Figure 15 and Appendix A). These hydration levels of the K-channel appear not to be influenced by the protonation state of the D-channel. An excess proton in the middle of the K-channel, i.e., either on K362 (models *10) or close to it (models *00d) results in almost comparably high numbers of water molecules in the K-channel, enough to bridge the lower part and upper part of the K-channel (see Figure 15 and Appendix A). Moreover, models in which the $H_{3}O^{+}$ ion is located just above S365 (models *00c) exhibit K-channel hydration that is not quite as high as in those models with the excess proton being located higher in the channel (models *10 and *00d), but they are significantly more hydrated than models in which the $H_{3}O^{+}$ ion is located below S365 (*00b). Of note, there is a probability to observe a water molecule between S365 and K362 as soon as the $H_{3}O^{+}$ ion is located above S365 (see Figure 15 and Appendix A). Again, these observations hold irrespectively of the protonation state of the D-channel (see Table 2). As also previously observed, the volume of the K-channel changes with its protonation state, in contrast to that of the D-channel. The K-channel exhibits a larger volume in those models that have a larger number of water molecules in the K-channel (see Appendix A for distances between residues defining the channel width). But it is also interesting to note that models *00e and *00f, in which the excess proton is located at the top of the channel but whose hydration level is rather low, exhibits a larger K-channel volume than models in which the excess proton is located in the lower part of the K-channel (models *01, *00a, and *00b) or not present at all (models *00). There is no impact of D-channel protonation on K-channel volume, however.

### 2.4. Hydrogen Bonds

Models with protonated E286 (models 01* and 11*) show a probability of presenting hydrogen bonds between N121 and N139 of about one, whereas models with charged E286 (models 00* and 10*) have a significantly lower probability of forming such a hydrogen bond (see Table 3). There are also some models with unprotonated E286 (0001, 0010, 0000e, 1010, 1000a, and 1000d) that show a high probability of presenting this hydrogen bond between the two asparagine residues, albeit with larger errors.

D132 exhibited a much higher probability of forming hydrogen bonds with water molecules inside the channel when it is charged than when it is protonated (see Figure 16). Accordingly, the lifetimes of hydrogen bonds between D132 and water molecules are also affected by D132’s protonation state, as can be seen from the very short-lived (i.e., below our time resolution of 2 ps) hydrogen bonds of protonated D132, with lifetimes of ∼30 ps to 50 ps (see Appendix A). Likewise, the protonation of E286 has a severe effect on the probabilities of hydrogen bonds between E286 and water molecules. Models with unprotonated E286 show a threefold higher probability of presenting such hydrogen bonds. This is clearly due to the also much higher number of water molecules inside the D-channel, which are mainly located in the upper part, i.e., close to E286. The lifetimes of hydrogen bonds between E286 and water molecules could not be determined for charged (unprotonated) E286 due to too-long lifetimes and/or too-large errors. In contrast, the lifetimes of hydrogen bonds between protonated E286 and water molecules vary between ∼50 ps and 70 ps.

The protonation state of E286 also affected the probability of finding hydrogen bonds between N121 or N139 and water molecules in the D-channel, but in a different manner. For N139, there was generally a higher probability to form hydrogen bonds with water molecules when E286 was protonated than when it was not. N121 had a trend towards a lower probability of forming hydrogen bonds with water molecules when E286 was protonated (see Figure 16). This was also reflected in the lifetimes of hydrogen bonds between N121 and water molecules, which were generally shorter by a factor of almost two when E286 was protonated than when it was unprotonated (models 01* and 11* compared with models 00* and 10*; see Appendix A). For H26, whose protonation state was neutral in all molecules, no clear dependence of the probability of forming hydrogen bonds with water molecules on the protonation state of the other channel residues, neither in the D- nor in the K-channel, could be observed (see Figure 16).

No significant influence of the K-channel protonation on the probability or lifetimes of hydrogen bonds between water molecules and D-channel residues could be observed (see Figure 16 and Appendix A). Similarly, most probabilities of finding hydrogen bonds between protein residues in the K-channel and water molecules are not affected by the protonation state of the D-channel. In contrast, the position of the proton inside the K-channel has an impact on the probability of finding hydrogen bonds between water molecules and protein residues in the K-channel (see Figure 17).

For T359 and S365, due to the fixed position of the $H_{3}O^{+}$ ion close to the respective residue (see Figure 5), hydrogen bonds can be formed with the $H_{3}O^{+}$ ion instead of water molecules. This effect is less pronounced for Y288, since due to the fixation of Y288’s Cα atom in addition to the fixed $H_{3}O^{+}$ oxygen atom position, the space between the $H_{3}O^{+}$ ion and the hydroxyl group of Y288 still allows hydrogen bonds to be formed with water molecules. Similar to E286 in the D-channel, E101 exhibits low probabilities of forming hydrogen bonds with water molecules when it is protonated (models *01 and *11). Moreover, a $H_{3}O^{+}$ ion in position a (close to E101 but not to S365) and then unprotonated E101 also results in a low probability of finding hydrogen bonds between E101 and water molecules (see Figure 17).

The probability of finding hydrogen bonds between K362 and water molecules is clearly higher when K362 is protonated than when it is neutral (see Figure 17). This is only partially due to the third proton at NZ being able to form another hydrogen bond, as can be seen from models 0011 and 1011, in which E101 is also protonated. These models exhibit a lower K-channel hydration level and with that also a lower probability of presenting hydrogen bonds with K362 (or any other residue) than *10 models with protonated K362 but unprotonated E101.

This effect of the hydration level is supported by the probabilities of hydrogen bonds between the $H_{3}O^{+}$ ion and water molecules (see Table 4), which are also higher in models with higher hydration levels (models with the $H_{3}O^{+}$ ion in position c or d). It is interesting to note that Y288 also exhibs a high probability of forming hydrogen bonds with water when the $H_{3}O^{+}$ ion is in position c. Hydrogen bonds between Y288 and water molecules also have the longest lifetimes in models with the $H_{3}O^{+}$ ion just above S365 (position c), except for model 1100d (with the $H_{3}O^{+}$ ion being placed next to K362), which exhibits longer hydrogen-bond lifetimes between Y288 and water molecules than model 1100c (see Appendix A).

Hydrogen-bond connections in the D-channel are present irrespective of the protonation model, and only the strengths varies. One remarkable effect is stronger hydrogen-bond connections between E286 and N121 with charged E286 as opposed to protonated E286 (see Figure 18 and Appendix A). D132 protonation results in less probable hydrogen-bond connections between D132 and H26, and between D132 and N139. For both acids (D132 and E286), showing lower probability to form hydrogen-bond connections in their protonated form is in agreement with the lower probability to form hydrogen bonds between these residues and water. In contrast, residues N121 and N139 of the asparagine gate, for which the probability to form hydrogen bonds with water changes with E286 protonation, are hydrogen-bond-connected with similar strength in all the models (see Figure 18 and Appendix A).

Residues in the K-channel of CcO exhibit a different pattern of hydrogen-bond connections, depending on the location of the proton in the K-channel. If the proton is located below S365 (models *00a and *00b), the hydrogen-bond connection network is split into an upper half and a lower half (see Figure 19 and Appendix A). One of the models with the $H_{3}O^{+}$ ion below S365 (1100b), however, exhibits weak connections between the lower part of the K-channel and K362 (see Figure 19).

Once, the proton has passed S365 (models *00c to *00f and models *10), both parts of the K-channel are connected by hydrogen bonds, unless E101 is also protonated (models *11), in which case, the connections in and to the lower part that are present in the *10 models are not observed. Model 0111 is an exception here, also showing hydrogen-bond connections between K362 and S362 or E101, respectively (see Figure 19).

Once the proton has arrived at the top of the channel, i.e., close to Y288 (models *00f), the hydrogen-bonded networks are again split into an upper part and a lower part (see Appendix A). This partitioning could also be observed in models 0000e and 1100e, in which the $H_{3}O^{+}$ ion is close to T359 (see Figure 19).

## 3. Discussion

According to the literature, the order of transfer events of electron, pumped proton, and “chemical” proton is still unclear. Electron transfer to the BNC may take place prior to the transfer of the proton to be pumped, or both transfers are fully coupled. There is also controversy on whether this electron transfer requires the “chemical” proton to be already at K362 or not [9,11,15].

In our simulations, the electron is already at the BNC, and the (unknown) proton loading site is not yet occupied by a proton. Rather, this proton is still somewhere in the D-channel (models 10*, 01*, and 11*) or has not yet entered the D-channel (models 00*). Therefore, our models cannot fully reflect the scenario after proton transport through the D-channel to the proton loading site and before proton transport through the K-channel. Our models rather assume that electron transfer to the BNC is before proton transfer to the proton loading site and has just occurred.

According to our data, both channels show individual regulation according to their own protonation state. The D-channel has a higher hydration level and is likely more proton conductive when E286 is not protonated. The persistent hydrogen bond between the two residues of the asparagine gate, N121 and N139, in models with protonated E286 suggests that a “closed” gate prevents water molecules from entering the D-channel. However, also in some models with unprotonated E286, there is a probability of finding a hydrogen bond between N121 and N139, corresponding to a closed asparagine gate. Yet, in those models, the number of water molecules in the D-channel is high. This leaves the possibility of water molecules having entered the D-channel by avoiding the asparagine gate or of the asparagine gate having opened (from the closed state in the crystal structure) and closed again during the equilibration phase of the simulation. In the latter case, i.e., with frequent opening and closing, the asparagine gate would not have a regulating role. However, charged E286 seems to exert higher attraction on water molecules than protonated E286, which is supported by the higher number of hydrogen bonds between E286 and water molecules when this residue is charged. Note that a similar effect was observed for D132 and E101, i.e., these residues show more hydrogen bonds with water molecules when unprotonated. This implies that the protonation state of E286 would regulate the likelihood of proton transport through the D-channel by attracting (or not) sufficient water molecules for proton transfer via hydrogen-bond connections. At least the hydrogen-bond connection between E286 and N121 is indeed more probable when E286 is not (yet) protonated.

The K-channel assumes its proton conductivity via the excess proton moving with its own hydration shell (see also [6]). With the excess proton located below S365, the hydration level of the K-channel is too low to render proton transport likely. This is in agreement with previous proposals of K362 protonation being the rate-determining step in proton transport through the K-channel [16]. From K362 to Y288, proton transport via hydrogen-bond connections is likely and even more so with K362 in an “up” conformation, as also shown in [17].

E101 protonation, even together with protonated K362, leads to a low hydration level in the K-channel, but does not harm proton transfer from K362 to Y288. It hinders, however, back transfer to the lower part of the K-channel due to low hydration and consequently low hydrogen-bond connectivity in the lower part of the K-channel. Furthermore, our previous work [6] has shown that hydrogen-bond connectivity and the associated proton transfer probability in the K-channel favour proton transfer “upwards” as soon as K362 is protonated and in an “up” conformation [18].

With K362, E101, and E286 protonated (model 0111), however, hydration and hydrogen-bond connectivity in the lower part of the K-channel suggest a non-negligible, albeit low, probability of back transfer. Still, the transfer of a proton from K362 to the protonated entry of the K-channel seems unlikely.

The changes in the protonation state of the D-channel can in principle be “sensed” by the K-channel through electrostatic interactions with either the $H_{3}O^{+}$ ion or the titratable residues K362 and E101, and by Y288. Likewise, the D-channel is “informed” about the protonation state of the K-channel via electrostatic interactions. Since the D-channel appears not to respond to those differences in K-channel protonation, at least not in the quantities observed in this work, any signalling between the two channels would then have to be from the D-channel to the K-channel.

The protonation of the D-channel, however, has hardly any influence on the conformation of K-channel residues or on the hydration level and hydrogen-bond connections in the K-channel. Two noteworthy exceptions are models 0111 and 1100b, which show hydrogen-bond connections between the upper part and lower part of the K-channel that are not found in the other models with the same K-channel protonation. In both cases, there are (more) water molecules in the K-channel, which make these hydrogen-bond connections possible (see Figure 15 and Appendix A). It is unclear whether these observations can be related to CcO function, although it is tempting to consider model 1100b as a state in which the D-channel has transported its protons, albeit not fully, to the proton loading site. In this state, the K-channel can now become active and the likely rate-determining transition to a protonation state with the proton past S365 and further up to K362 (such as models 1100c and 1110) can take place. If, however, a completed proton passage through the D-channel is required to “activate” the K-channel, model 0100b should show an effect similar to that of model 1100b, but this is not the case. Moreover, the probability of finding a hydrogen-bond connection to K362 is also rather weak in model 1100b (∼0.1). There is also no obvious explanation why only in model 1100b, water molecules are more probable at the height of S365 and slightly above, thereby making hydrogen-bond connections and eventually proton passage through the K-channel possible.

The apparent independence of the two proton-conducting channels of CcO, as manifested in the analysed properties (conformational preference for key protein residues, hydration level, and hydrogen-bond connections, with the latter being perhaps most important for proton transport), does not rule out that the activity of the two channels is regulated by proton transfer events. Our data suggest, however, that such regulation is not achieved via direct communication between the two channels (“sensing” their respective protonation states). Another possible way is indirect communication not, or not solely, via the location of the excess proton but rather the injected electron and the associated regulation of proton transfer events.

After the injection of the first electron in the reductive phase and its transfer from CuA to heme a, a proton is transported through the D-channel to the proton loading site (somewhere above the BNC). This is coupled to (partial) electron transfer from heme a to the BNC. This electron transfer is denoted here as partial, since spectroscopic data suggest only 60% of the electron to be transferred from heme a to the BNC [10,12].

The negative charge at the BNC then triggers proton transfer through the K-channel. The communication of the two channels is thus not performed directly through their protonation states. Rather, the first proton transfer through the D-channel is likely independent of the K-channel’s protonation state. However, the accomplishment of proton delivery through the D-channel (to the proton loading site) and the electron transfer coupled therewith are signalled to the K-channel.

A mechanism in which the pumped proton passes the D-channel before the electron reaches the BNC has indeed been previously proposed [10]. This proposal is also in agreement with the effect of K362M mutation on the rate of reduction of heme a3, as measured using time-resolved optical spectroscopy and EPR [12]. In that work, even with an inactive K-channel, one electron could reduce the BNC but was found to be shared between heme a3 and CuB in the *Rh. sphaeroides* oxidase. Complete proton pumping, however, requires expelling the proton from the proton loading site, which is only achieved with a proton arriving at the BNC, delivered through the K-channel.

The notion of K362 being already protonated, required for and facilitating electron transfer to the BNC, as mentioned in [11], is not supported by our data. All simulations with protonated K362 show a high hydration level of the K-channel that is in contradiction with the few water molecules observed in the crystal structure [8]. Even in the models in which E101 is also protonated (models *11), which exhibit a significantly lower hydration level, there are enough water molecules present in the upper half of the channel leading to hydrogen-bond connectivity that allows proton transfer from K362 to Y288. All that could prevent this proton transfer from happening before proton transport through the D-channel is the fast electron transfer to CuB and a proton transfer to the proton loading site that is faster than the proton transfer from K362 to the BNC. Models in which the proton transfer through the D-channel and further to the proton loading site via water molecules and hydrogen-bond connections above E286 is much faster have indeed been proposed [19,20,21]. However, simulations of proton transfer from E286 to the putative proton loading site, albeit in the oxidative phase, show barriers of about 10 kcal/mol [22,23] and are thus comparable to that of proton transfer from K362 to Y288, which has been calculated to be about 11 kcal/mol [17].

K-channel protonation, at least in the upper part of the channel (at K362 or at position e or f), can be anticipated to be less likely without the electron at CuB. Though not fully comparable, simulations of this proton transfer found indeed a higher barrier in the oxidised Pr state than in the reduced O→E state [17]. One can envisage, however, that protons can enter the K-channel up to position b (just below S365) without further attraction by an electron at the BNC.

The (partial) electron transfer to the BNC coupled to or followed by proton transfer to the proton loading site can thus be regarded as the “go” signal for the K-channel. Only with the electron there, the proton affinity of the BNC is high enough to enable the proton and its accompanying water molecules to travel beyond S365. Once this residue has been passed, further hydration and proton passage to K362 and all the way up to Y288 become feasible.

Since there is no pronounced effect of D-channel protonation on the K-channel, the K-channel is “insensitive” to the proton to be pumped or a proton to re-load E286 while making its way through the D-channel. Any “sensitivity” that regulates K-channel activity must, therefore, be towards the redox state of the BNC (and perhaps to a proton at the proton loading site, since that would counteract the negative charge at the BNC to some extent).

## 4. Materials and Methods

### 4.1. Molecular Dynamics Simulations

#### Model Setup

The model setup was the same as that previously used for our molecular dynamics simulations of the Pr→ F redox state [24] and of the O→E state [6,18] and is repeated here for convenience. The redox state was modelled as described in ref. [16], with a hydroxide ion bound to the Fe(III) of the heme, a water molecule coordinated to the Cu(I) of the BNC, and the terminal residue of the K-channel, Y288, in the deprotonated state (resembling state $O_{H}$ in Figure 1, but with Cu(I) to model the reduction by one electron). Y288 was also cross-linked to H284 (see [16]).

The protonation states of the D-channel were varied with protonated/unprotonated D132 and E286, respectively, resulting in four protonation possibilities of the D-channel. These were combined with different protonation states of the K-channel, varying the protonation of residues K362 and E101 in all four possible combinations.

Furthermore, we performed six additional sets of simulations in which an excess proton is bound to a water molecule in the K-channel, represented by a crystal water molecule converted to a hydronium ion, as also described in our previous work [6]. The respective positions of the hydronium ions are as follows (the values in parentheses are the distances between the oxygen atom of the $H_{3}O^{+}$ ions and the Cα atom of Y288 (in Å)):(a)Between H96 and E101 (23.4).(b)Between E101 and S365 (19.1).(c)Between S365 and K362 (15.5).(d)Just below K362 (12.8).(e)Between K362 and T359 (10.4).(f)Between T359 and Y288 (7.3).

In all these simulations, the oxygen atom of the hydronium ions was kept stationary, together with the Cα atom of Y288, so as to maintain the relative height of the hydronium ion in the channel. The hydrogen atoms of the hydronium ions were allowed to move so as to allow them to re-orientate according to the dynamic environment.

The resulting protonation possibilities are indicated by a binary code, where 0 means unprotonated and 1 means protonated, respectively. The first four digits have the same meaning as in our previous work [6,7,24], referring to D132, E286, K362, and E101, respectively. (Note that some analyses for the K-channel of the simulations with two residues of the D-channel represented as 01 (Asp132 unprotonated and E286 protonated) have already been reported in [6] and are reported in this work for comparison.) As an example, model 1010 refers to a model with protonated D132, unprotonated E286, protonated K362, and unprotonated E101.

Models with the proton being located on a water molecule (as hydronium ion) are labelled xy00z, where x and y are the protonation codes of D132 and E286, respectively, and z = “a” to “f” represent the positions of the hydronium ($H_{3}O^{+}$) ion. K362 and E101 are unprotonated in those models. For example, 1000b refers to a model in which D132 is protonated and the hydronium ion is placed between E101 and S365. The protonation models studied in this work and their protonation codes are listed in Table 5.

When we refer to a group of models, we represent the common protonation state (digits) with *. For example, models *b refers to all models with a $H_{3}O^{+}$ ion in position b, regardless of the protonation state of the protein residues (strictly speaking, of those in the D-channel). Accordingly, 01* refers to all models in which D132 is not protonated, E286 is protonated, and K362 and E101 can be in any protonation state, or K362 and E101 are unprotonated and a $H_{3}O^{+}$ is present in any of the positions a to f.

The core enzyme complex (subunits I and II) was embedded in a lipid bilayer of phosphatidylcholines and solvated in TIP3 water [25] (25,882 water molecules, ∼115,000 atoms in total). ${\mathrm{Na}}^{+}$ counter-ions were added by randomly substituting water oxygen atoms to neutralize the charge of the system. For protein residues, the CHARMM22 force field [26] was applied, while the parameters for the cofactors were based on quantum chemically derived atomic partial charges and optimized cofactor geometry by Woelke et al. [16], and the parameters for the hydronium ion were taken from Ref. [27]. The parameters for the lipid bilayer were obtained from the CHARMM36 extension for lipids [28]. Simulations were performed using periodic boundary conditions in a tetragonal box of sizes x = y = 96 Å and z = 124 Å. Long-range electrostatic interactions were treated using the Particle Mesh Ewald method [29] on a 96 × 96 × 128 charge grid. A non-bonded cut-off of 12 Å was applied. The short-range electrostatics and van der Waals interactions were truncated at 12 Å using a switch function starting at 10 Å.

**Table 5 ijms-24-10464-t005:** Protonation models of CcO.

Model	Protonated D-Channel Residue(s)	Protonated K-Channel Residue(s)
0000	-	-
0001	-	E101
0010	-	K362
0011	-	K362 and E101
0000a	-	$H_{3}O^{+}$ in position a
0000b	-	$H_{3}O^{+}$ in position b
0000c	-	$H_{3}O^{+}$ in position c
0000d	-	$H_{3}O^{+}$ in position d
0000e	-	$H_{3}O^{+}$ in position e
0000f	-	H${}_{3}O^{+}$ in position f
0100	E286	-
0101	E286	E101
0110	E286	K362
0111	E286	K362 and E101
0100a	E286	$H_{3}O^{+}$ in position a
0100b	E286	$H_{3}O^{+}$ in position b
0100c	E286	$H_{3}O^{+}$ in position c
0100d	E286	$H_{3}O^{+}$ in position d
0100e	E286	$H_{3}O^{+}$ in position e
0100f	E286	$H_{3}O^{+}$ in position f
1000	D132	-
1001	D132	E101
1010	D132	K362
1011	D132	K362 and E101
1000a	D132	$H_{3}O^{+}$ in position a
1000b	D132	$H_{3}O^{+}$ in position b
1000c	D132	$H_{3}O^{+}$ in position c
1000d	D132	$H_{3}O^{+}$ in position d
1000e	D132	$H_{3}O^{+}$ in position e
1000f	D132	$H_{3}O^{+}$ in position f
1100	D132 and E286	-
1101	D132 and E286	E101
1110	D132 and E286	K362
1111	D132 and E286	K362 and E101
1100a	D132 and E286	$H_{3}O^{+}$ in position a
1100b	D132 and E286	$H_{3}O^{+}$ in position b
1100c	D132 and E286	$H_{3}O^{+}$ in position c
1100d	D132 and E286	$H_{3}O^{+}$ in position d
1100e	D132 and E286	$H_{3}O^{+}$ in position e
1100f	D132 and E286	$H_{3}O^{+}$ in position f

### 4.2. Molecular Dynamics Simulations

After minimising the systems for 5000 steps using steepest descent and heating for 30 ps to 300 K, three stages of equilibration (with decreasing harmonic restraints on the solute atoms) were performed, in which the numbers of particles, pressure (1 bar), and temperature (300 K) were kept constant (NPT ensemble) for 75 ps. Pressure control was introduced using the Nosé–Hoover–Langevin piston with a decay period of 500 fs. Then, three replicas, started with different initial velocities, of 200 ns long NPT production runs with a 2 fs integration time step (using a velocity verlet integrator) were performed, and the coordinates were saved every 2 ps. All covalent bond lengths involving hydrogen atoms were fixed using the SHAKE algorithm [30].

These molecular dynamics (MD) simulations were carried out using the program NAMD2.14 [31].

### 4.3. Analysis

The first 40 ns of the simulation time was regarded as further equilibration, similar to our previous work [6,7,18,24], thus leaving the last 160 ns for analysis. All properties were separately evaluated for each of the three independent simulations and then averaged. The standard error from this mean was used for error estimates.

The analyses carried out in this work largely followed the same approaches used in our previous works [6,7,24] and are specified below.

All analyses were performed using our own python scripts and Java-written code, which uses the JGraph library [32]. Our python implementation for the calculation of hydrogen-bond lifetime is an adaption of the respective code in MDanalysis [33]. Molecule figures were generated with vmd [34], and all plots were generated with matplotlib in python [35], making use of the Networkx library [36] for representations of graphs.

#### 4.3.1. Conformational Analysis

Similar to our previous work [6], we analysed the side-chain conformations, i.e., dihedral angles $\chi_{1}$ to $\chi_{4}$ (where applicable), of protein residues D132, N121, N139, E286, Y288, T359, K362, S365, H96, and E101, as well as the distances between some of these residues. For distance analysis, we considered the polar atoms of the side chains, i.e., oxygen atoms in tyrosine, serine, and threonine residues, and the nitrogen atom Nζ in K362, respectively. For the two asparagine residues, N121 and N139, we analysed distances from the ND2 and OD1 atoms, respectively. For H26 and H96, we used atom Cϵ1, to which the two nitrogen atoms of the side chain are bound. For D132, we used the Cγ atom, and for glutamate residues E286 and E101, we used the Cδ atoms, to which the respective two oxygen atoms of the side chains are bound. We refer to an “up” conformation if a residue’s side chain points towards the P-side of the membrane (towards the BNC) and a “down” conformation when such a side chain points towards the N-side of the membrane.

Furthermore, the distances between the Cα atoms of P358 and A319, M316 and K362, P315 and E101, and P315 and S365 were analysed as a measure of the width of the K-channel. The former two were previously used in ref. [16], and the latter two were additionally used in [6] as a measure of the channel width in the lower region.

#### 4.3.2. Electrostatic Interactions

Electrostatic interactions were computed using NAMD2.14, including the full system and the same settings (parameters, cut-offs, and PME grid size) as in the MD simulations. Interactions between the following pairs of residues were computed: E286 and N139, and E286 and D132 in the D-channel; K362 and E101, and Y288 and K362 in the K-channel; and between E286 and Y288, E286 and K362, K362 and N139, and K362 and D132 as interactions across the two channels. Moreover, electrostatic interactions were calculated between the following residues and the $H_{3}O^{+}$ ion, if present: Y288, K362, E101, D132, N139, and E286.

#### 4.3.3. Channel Hydration

To identify water molecules that interact with the D- or K-channel, we defined the volume of the channels with a polyhedron. The corners of the polyhedron defining the D-channel were given by the centres of mass of the residues T24, Y33, F109, V110, N121, D132, P136, M138, N139, S142, S197, S200, S201, N207, F282, E286, and I289 of chain A. The centres of mass of the residues L227, L238, H284, V287, P315, G324, G327, A356, I363, S365, W366, and G398 of chain A and H96, L100, and E101 of chain B were used to define the K-channel polyhedron. For every time frame in the trajectory, the volume-maximised polyhedron was calculated (see Figure 20). Each water molecule, determined by the position of its oxygen atom, that hit the polyhedron volume at least once in the analysed simulations time was considered for further evaluations.

**Figure 20 ijms-24-10464-f020:**
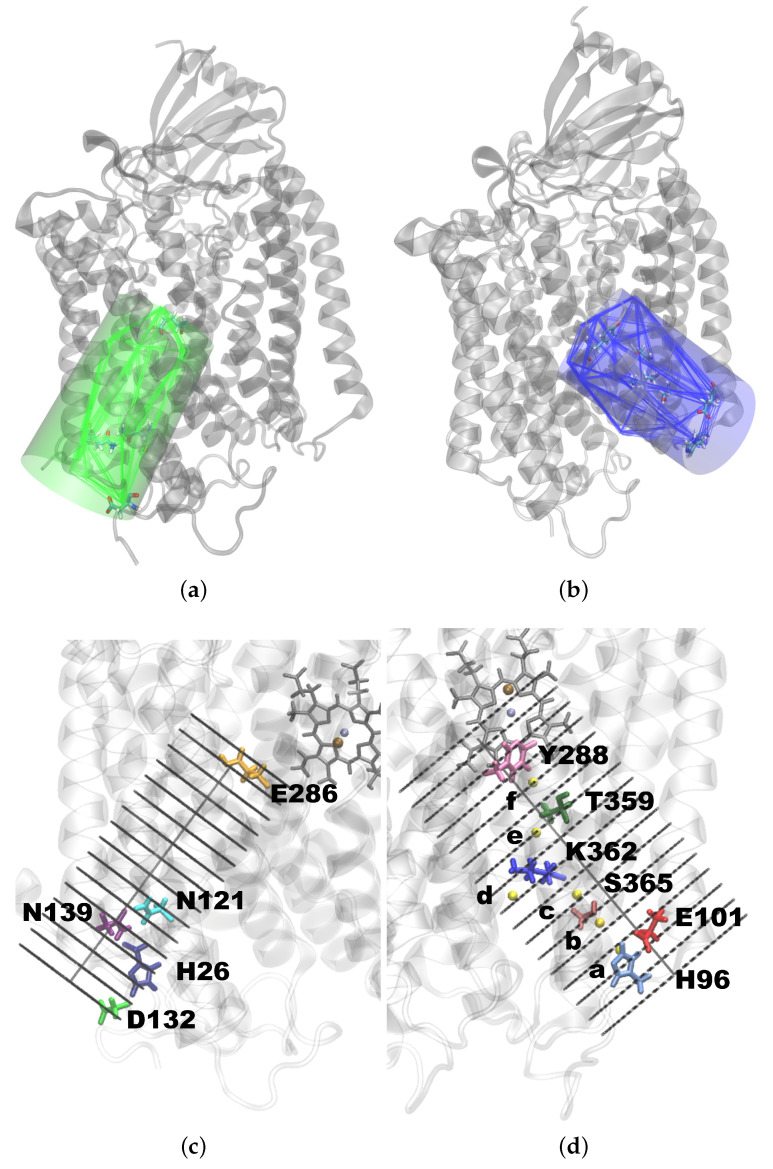
Polyhedron of (**a**) the D-channel and (**b**) the K-channel drawn every 16 ns enveloped using a cylinder. Partitioning of the enveloping cylinder of (**c**) the D-channel and (**d**) the K-channel into slices. The protein is shown in cartoon, and important protein residues, in stick representation, coloured as follows: H26: dark blue; N121: cyan; D132: lime; N139: purple; E286: orange; Y288: pink; T359: green; K362: blue; S365 brown; H96: light blue; E101: red. Possible locations of the $H_{3}O^{+}$ ion (a–f) are indicated by yellow spheres.

For spatial visualisations within the channel, the polyhedron was enveloped using a cylinder of radius 9.5 Å such that all water molecules that were identified to be within the polyhedron were also captured inside the cylinder. To construct the cylinder, the frame closest to the average atom positions (representing a “median structure”) was used. This frame was separately calculated for each trajectory. The cylinder axis was formed between the centres of mass of the $C_{\alpha }$ atoms of the lowermost and uppermost residues. For the proper coverage of the entrance and exit of the channel, the axis was elongated by 10%. Its resulting maximum height was 30 Å. Along the axis, the cylinder height was partitioned into 15 slices of 2 Å in thickness each. The thickness of a slice was chosen to be larger than the largest distance in a (TIP3P) water molecule (the H–H distance is ∼1.5 Å) but by about the same amount smaller than the H⋯O distance in a hydrogen bond (∼2.5 Å) to a neighbouring water molecule.

#### 4.3.4. Hydrogen-Bond Probabilities and Hydrogen-Bond Lifetimes

We analysed hydrogen bonds between D-channel residues N121 and N139, and those between water molecules and protein residues: D-channel residues H26, D132, N121, N139, and E286; K-channel residues Y288, T359, K362, S365, H96, and E101 in the respective channels. The protocol was the same as that in [6] and is repeated here for convenience and augmented by the analysis of the D-channel. A hydrogen bond is defined by the geometric criterion of a donor–acceptor distance not larger than 3.5 Å and a maximal deviation from a linear donor–hydrogen–acceptor (D-H⋯A) angle of 35 degrees.

Hydrogen-bond lifetimes were calculated between important protein residues (H26, N121, D132, N139, E286, Y288, T359, K362, S365, H96, E101, and the $H_{3}O^{+}$ ion, if present) and water molecules with the time auto-correlation function using the following (Equation (Equation 1)):(1)$$C\left(\tau \right)=\langle \frac{\sum h\left(t_{0}\right)h\left(t_{0}+\tau \right)}{\sum h{\left(t_{0}\right)}^{2}}\rangle$$
where *h* is a binary measure of whether a hydrogen bond exists, h=1, or not, h=0, irrespective of the identity of the water molecule; $h(t_{0})=1$ indicates a hydrogen bond at time $t_{0}$; and $h(t_{0}+\tau )=1$ indicates that the protein residue remains hydrogen-bonded to a water molecule throughout the period $t_{0}$ to $t_{0}+\tau$. Note that the individual water molecule, as well as the individual donor, acceptor, and hydrogen atoms, can change. The hydrogen-bond lifetime is obtained by fitting the time auto-correlation curve with a bi-exponential function:(2)$$C\left(t\right)=A\cdot exp\left(-t/\tau_{1}\right)+B\cdot exp\left(-t/\tau_{2}\right)$$
where $\tau_{1}$ and $\tau_{2}$ represent two time constants, one corresponding to a short-timescale process and the other corresponding to a longer-timescale process. Amplitudes *A* and *B*, which add up to 1, represent the respective weights of the short- and longer-timescale processes in the overall auto-correlation curve [33]. The reported hydrogen-bond lifetimes are the weighted sums of the time constants of the two processes: $\tau =A\cdot \tau_{1}+B\cdot \tau_{2}$.

We evaluated the auto-correlation up to maximal lag time τ=100 ps. To improve statistics, multiple time origins, $t_{0}$, were used in the calculation, and the average was taken over all time origins. The period between time origins, $t_{0}$, was chosen as 120 ps such that the individual auto-correlation sequences did not overlap.

Hydrogen-bond connections between protein residues inside the D-channel (i.e., among H26, N121, D132, N139, and E286) or inside the K-channel (i.e., Y288, T359, K362, S365, H96, and E101), and, if present, the $H_{3}O^{+}$ ion, could be formed directly or via water molecules. To find the probabilities for such water-mediated hydrogen-bond connections, for each time frame of the simulation, a graph was set up in which the protein residues and water molecules represented the nodes and the edges were represented by existing hydrogen bonds in a pair of residues in that frame. On that graph, a shortest-path search using Dijkstra’s algorithm [37] between two nodes representing protein residues was performed. If such a path existed, the hydrogen-bond connection between the two residues considered was counted. Hydrogen-bond connections were separately computed for the D-channel and K-channel.

## 5. Conclusions

The hydration level and consequently the interactions between protein residues and water, as well as water-mediated interactions between protein residues, in the two proton-conducting channels of Cytochrome c Oxidase clearly depend on the protonation state of the respective channel. However, no significant effect of the protonation state of one channel on these properties of the other channel could be observed. The regulation of proton transport through the two channels in the correct order must, therefore, be achieved via other means, such as electron transfer to the BNC. Since this is coupled to the proton transport through the D-channel to the (unknown) proton loading site, this can be understood as (indirect) communication from the D-channel to the K-channel.

## Figures and Tables

**Figure 1 ijms-24-10464-f001:**
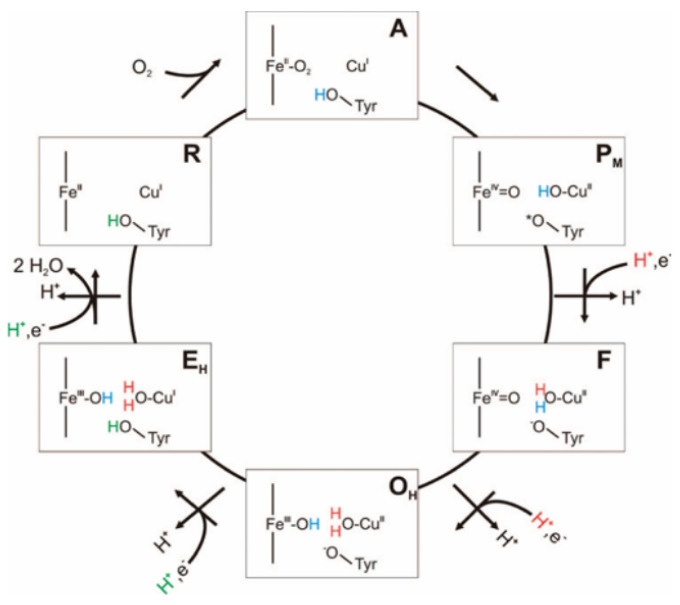
Redox cycle in Cytochrome c Oxidase. Figure taken from [6].

**Figure 2 ijms-24-10464-f002:**
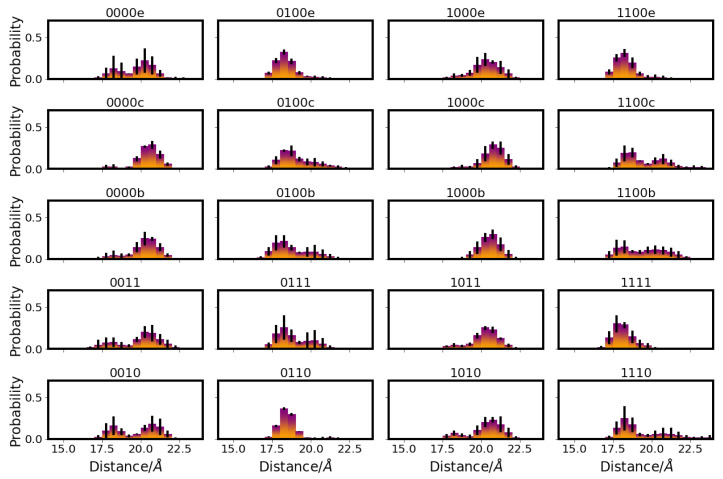
Distribution of distances between E286 and N139 in different protonation models of CcO. For the other models, see Appendix A. The different models are labelled according to the protonation states of residues D132, E286, K362, and E101, where “0” means unprotonated and “1” means protonated. Letters a–f refer to a $H_{3}O^{+}$ ion in the K-channel at different heights (see Table 5 in “Materials and Methods”).

**Figure 3 ijms-24-10464-f003:**
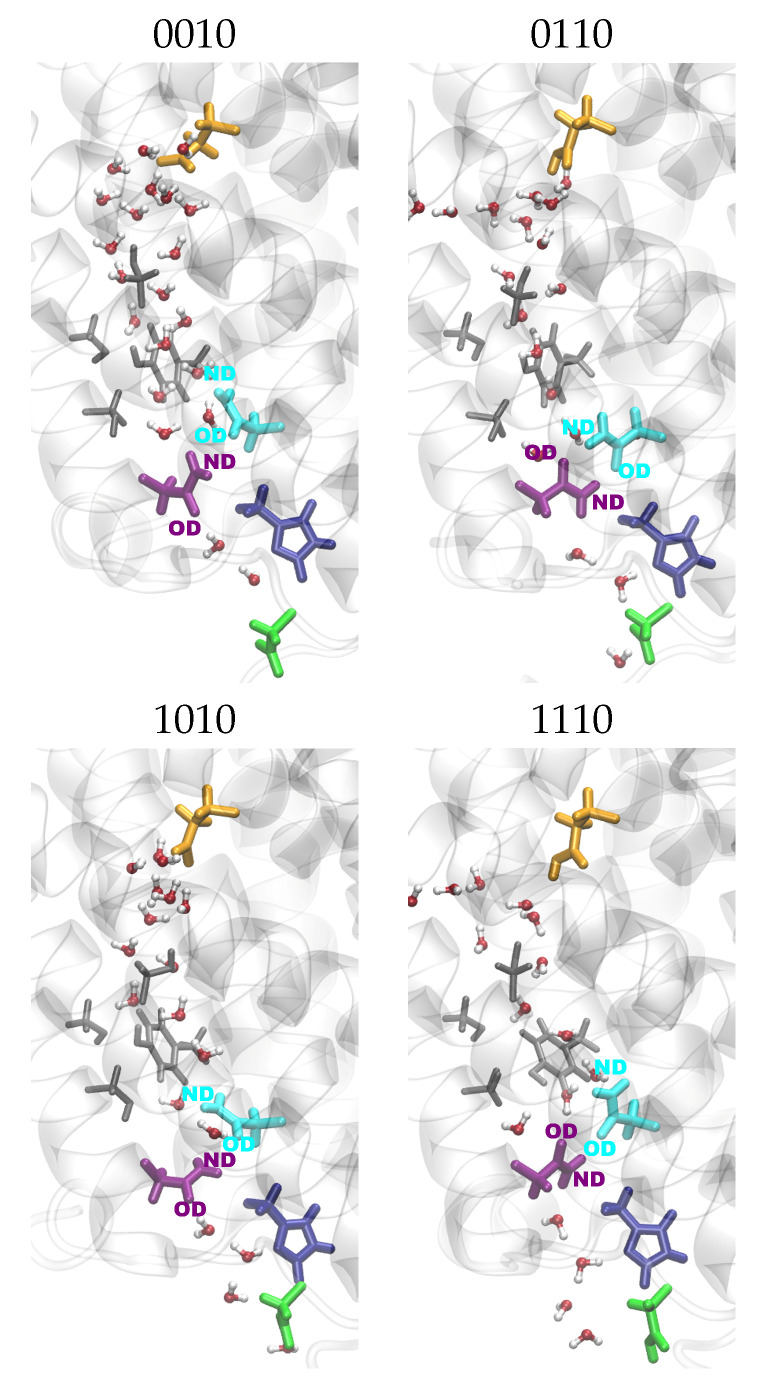
Snapshots of the D-channel for models with varying D-channel protonation. The different models are labelled according to the protonation states of residues D132, E286, K362, and E101, where “0” means unprotonated and “1” means protonated (see Table 5 in “Materials and Methods”). Important residues are shown in colour (dark blue: H26; cyan: N121; lime: D132; purple: N139; orange: E286).

**Figure 4 ijms-24-10464-f004:**
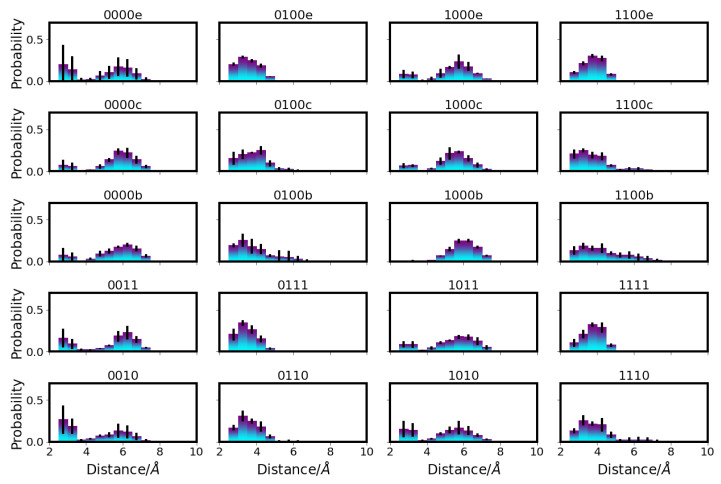
Distribution of distances between N121 and N139 in different protonation models of CcO. For the other models, see Appendix A. The different models are labelled according to the protonation states of residues D132, E286, K362, and E101, where “0” means unprotonated and “1” means protonated. Letters a–f refer to a $H_{3}O^{+}$ ion in the K-channel at different heights (see also Table 5 in “Materials and Methods”).

**Figure 5 ijms-24-10464-f005:**
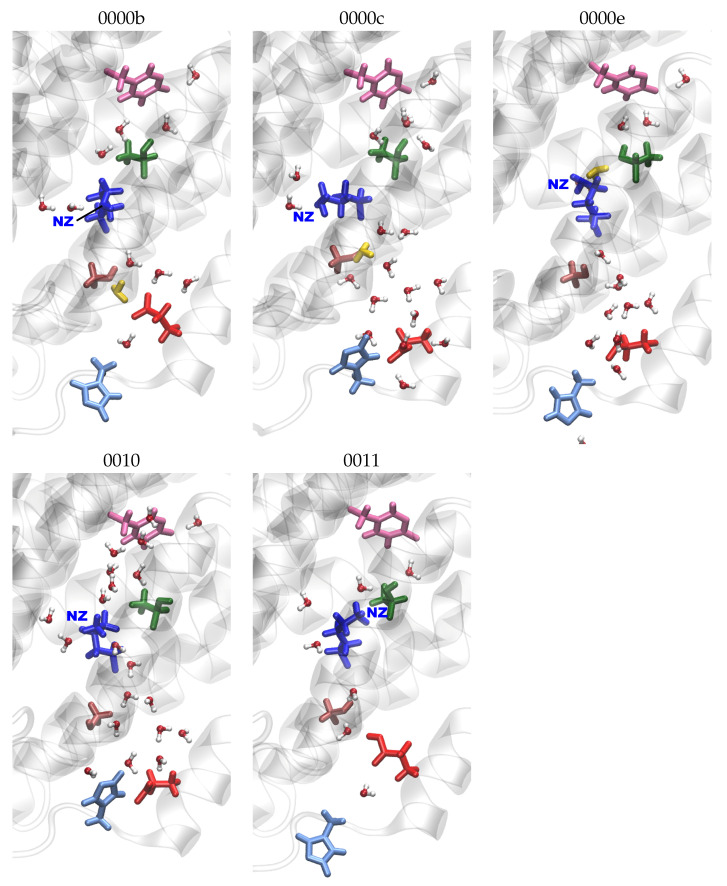
Snapshots of the K-channel for models with varying K-channel protonation. The different models are labelled according to the protonation states of residues D132, E286, K362, and E101, where “0” means unprotonated and “1” means protonated (see Table 5 in “Materials and Methods”). Letters a–f refer to a $H_{3}O^{+}$ ion in the K-channel at different heights (see also Figure 20d) in “Materials and Methods”). Important residues are shown in colour (pink: Y288; dark green: T359; blue: K362; brown: S365; light blue: H96; red: E101; yellow: $H_{3}O^{+}$ ion).

**Figure 6 ijms-24-10464-f006:**
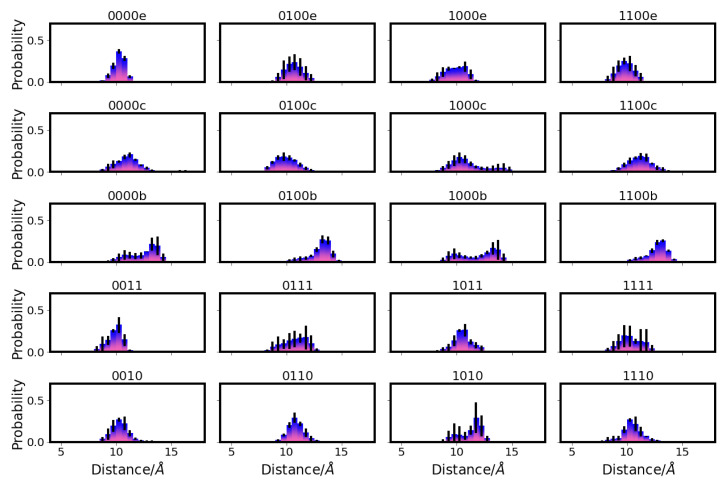
Distribution of distances between Y288 and K362 in different protonation models of CcO. For the other models, see Appendix A. The different models are labelled according to the protonation states of residues D132, E286, K362, and E101, where “0” means unprotonated and “1” means protonated (see Table 5 in “Materials and Methods”). Letters a–f refer to a $H_{3}O^{+}$ ion in the K-channel at different heights (see also Figure 20d in “Materials and Methods”).

**Figure 7 ijms-24-10464-f007:**
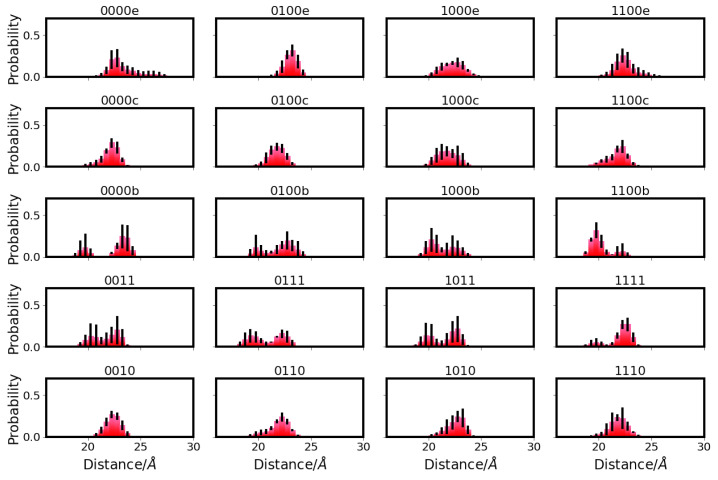
Distribution of distances between E101 and Y288 in different protonation models of CcO. For the other models, see Appendix A. The different models are labelled according to the protonation states of residues D132, E286, K362, and E101, where “0” means unprotonated and “1” means protonated (see Table 5 in “Materials and Methods”). Letters a–f refer to a $H_{3}O^{+}$ ion in the K-channel at different heights (see also Figure 20d in “Materials and Methods”).

**Figure 8 ijms-24-10464-f008:**
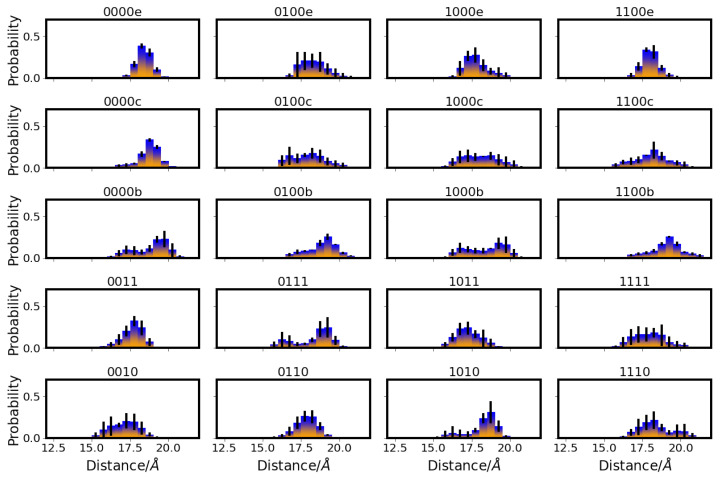
Distribution of distances between E286 and K362 in different protonation models of CcO. For the other models, see Appendix A. The different models are labelled according to the protonation states of residues D132, E286, K362, and E101, where “0” means unprotonated and “1” means protonated (see Table 5 in “Materials and Methods”). Letters a–f refer to a $H_{3}O^{+}$ ion in the K-channel at different heights (see also Figure 20d in “Materials and Methods”).

**Figure 9 ijms-24-10464-f009:**
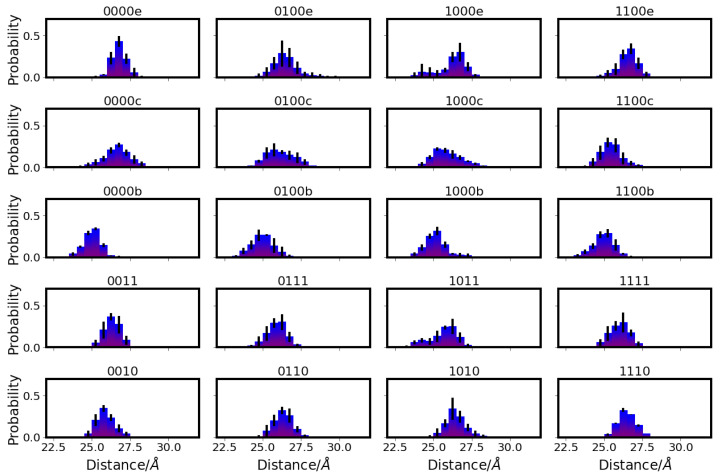
Distribution of distances between N139 and K362 in different protonation models of CcO. For the other models, see Appendix A. The different models are labelled according to the protonation states of residues D132, E286, K362, and E101, where “0” means unprotonated and “1” means protonated (see Table 5 in “Materials and Methods”). Letters a–f refer to a $H_{3}O^{+}$ ion in the K-channel at different heights (see also Figure 20d in “Materials and Methods”).

**Figure 10 ijms-24-10464-f010:**
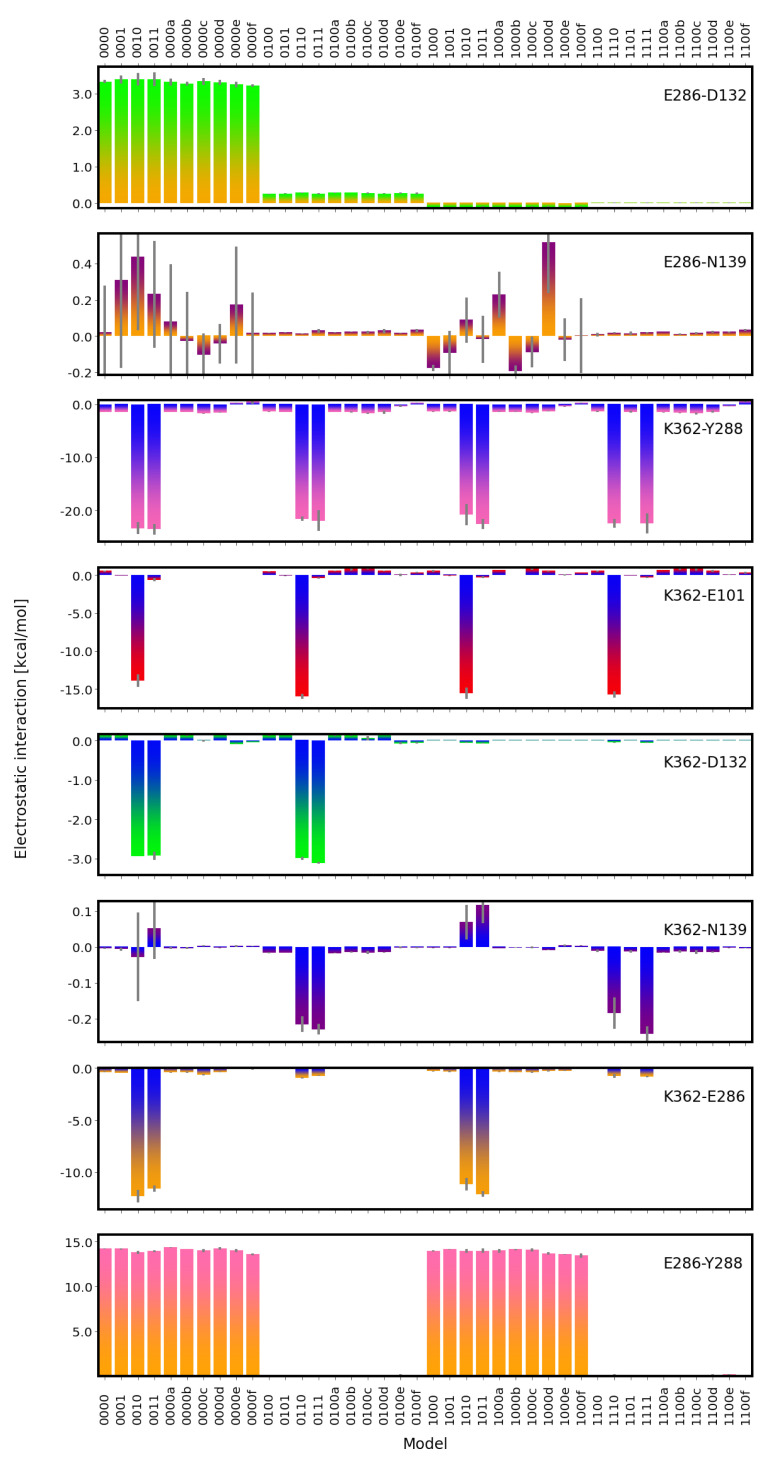
Electrostatic interactions between important residues in the D-channel and the K-channel of CcO. The different models are labelled according to the protonation states of residues D132, E286, K362, and E101, where “0” means unprotonated and “1” means protonated (see Table 5 in “Materials and Methods”). Letters a–f refer to a $H_{3}O^{+}$ ion in the K-channel at different heights (see also Figure 20d in “Materials and Methods”).

**Figure 11 ijms-24-10464-f011:**
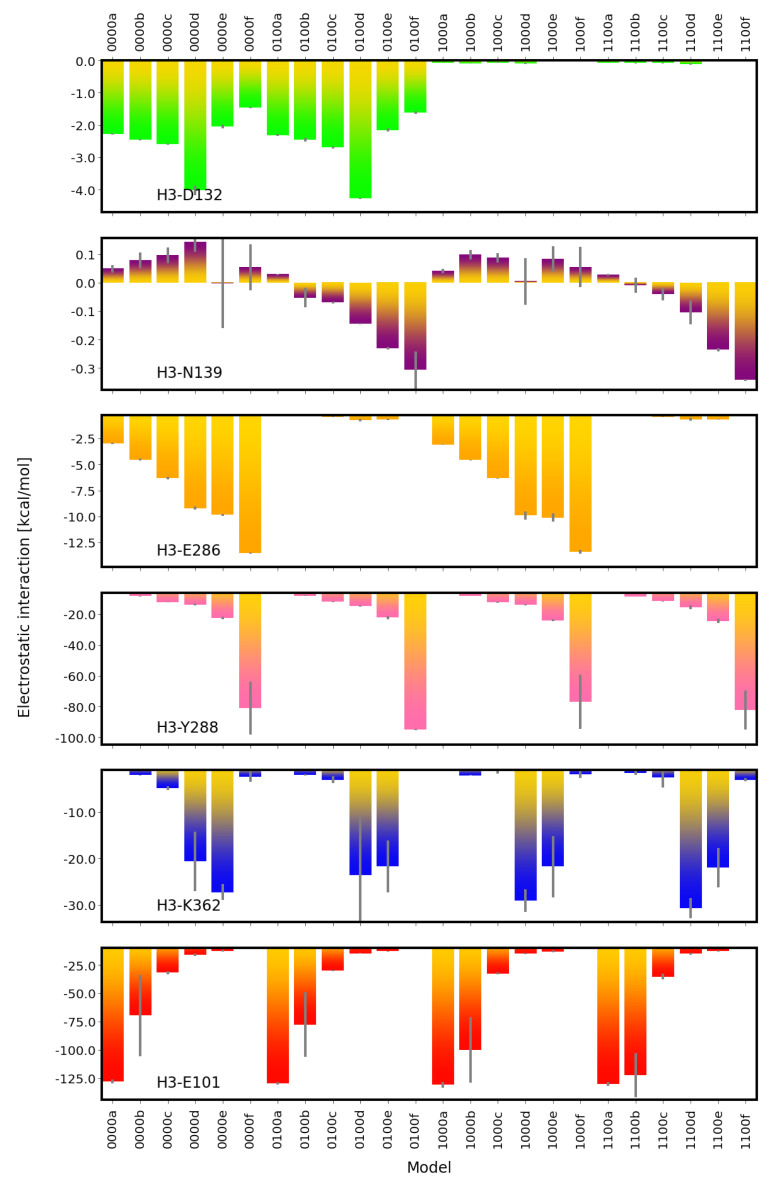
Electrostatic interactions between the $H_{3}O^{+}$ ion and important residues in the D-channel and the K-channel of CcO. Distances between the $H_{3}O^{+}$ ion and the residues are reported in Appendix A. The different models are labelled according to the protonation states of residues D132, E286, K362, and E101, where “0” means unprotonated and “1” means protonated (see Table 5 in “Materials and Methods”). Letters a–f refer to a $H_{3}O^{+}$ ion in the K-channel at different heights (see also Figure 20d in “Materials and Methods”).

**Figure 12 ijms-24-10464-f012:**
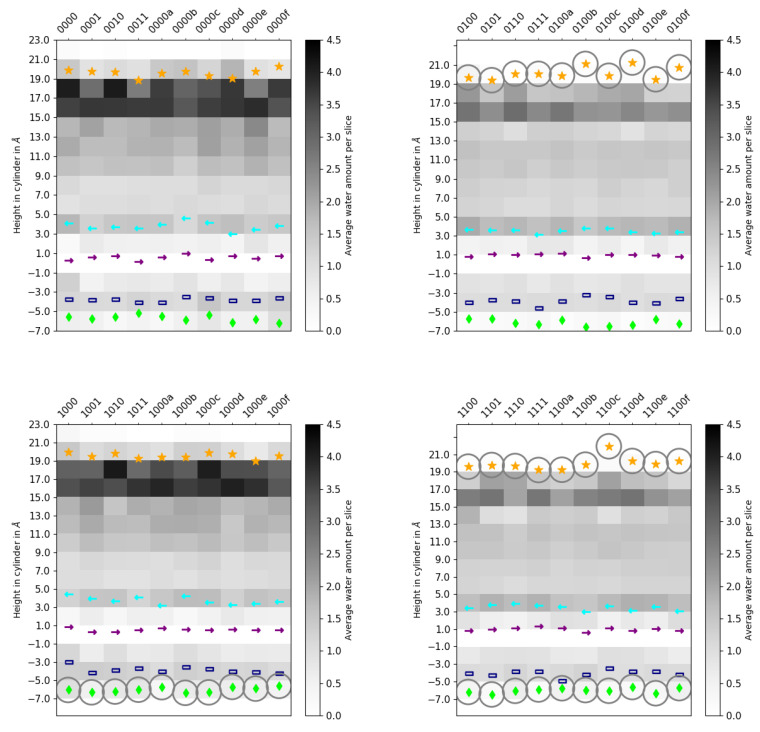
Number of water molecules in the D-channel, discretised by cylinder height for different protonation models of the D-channel of CcO. (Corresponding mobilities of the water molecules are reported in Appendix A.) The different models are labelled according to the protonation states of residues D132, E286, K362, and E101, where “0” means unprotonated and “1” means protonated (see Table 5 in “Materials and Methods”). The symbols mark the average height of protein residues H26 (dark blue rectangle 
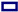
), N121 (cyan arrow—left 

), D132 (lime diamond 

), N139 (purple arrow—right 

), and E286 (orange star 
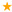
). Residues with an excess proton are marked by a grey circle around the symbols representing the respective residues.

**Figure 13 ijms-24-10464-f013:**
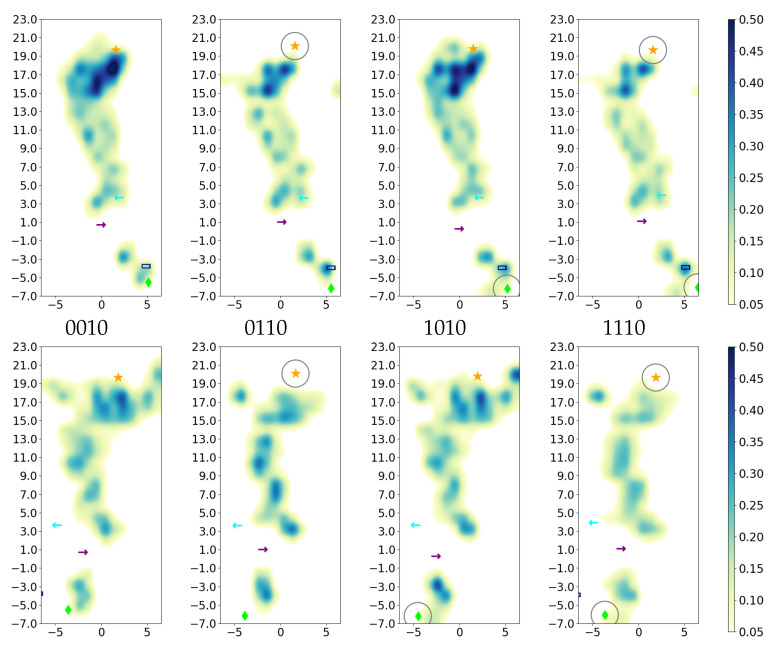
Projection of water occupancy in the D-channel of CcO for varying D-channel protonation. The different models are labelled according to the protonation states of residues D132, E286, K362, and E101, where “0” means unprotonated and “1” means protonated (see Table 5 in “Materials and Methods”). (For the projections of all models, see Appendix A.) In the K-channel, K362 is protonated (*10 models). The lower row shows a view 90 degrees rotated from that of the upper row. The symbols mark the average height of protein residues H26 (dark blue rectangle 
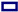
), N121 (cyan arrow—left 

), D132 (lime diamond 

), N139 (purple arrow—right 

), and E286 (orange star 
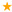
). Residues with an excess proton are marked by a grey circle around the symbols representing the respective residues.

**Figure 14 ijms-24-10464-f014:**
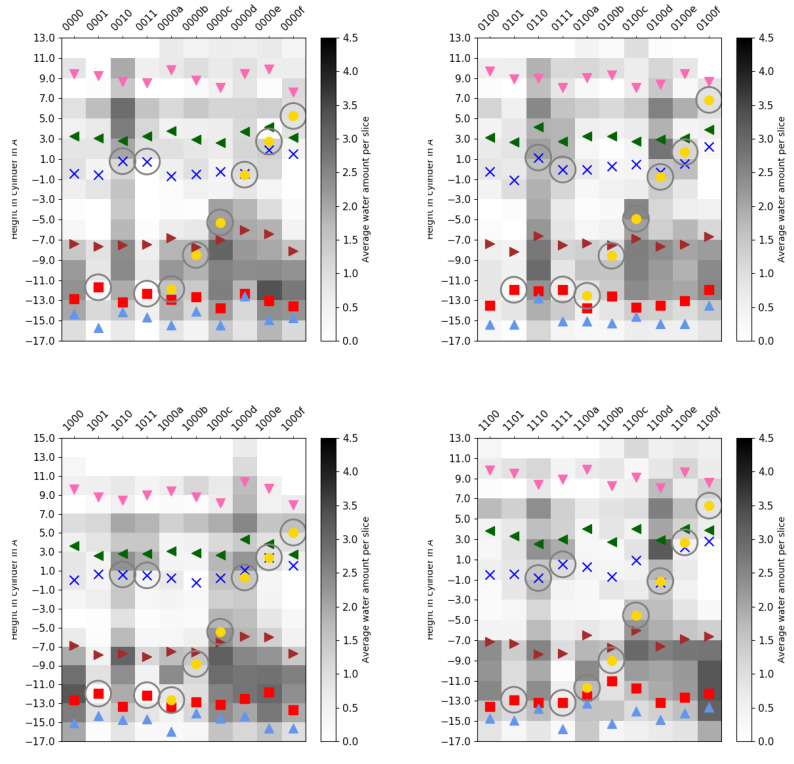
Number of water molecules in the K-channel, discretised by cylinder height for different protonation models of the K-channel of CcO. (Corresponding mobilities of the water molecules are reported in Appendix A.) The different models are labelled according to the protonation states of residues D132, E286, K362, and E101, where “0” means unprotonated and “1” means protonated (see Table 5 in “Materials and Methods”). Letters a–f refer to a $H_{3}O^{+}$ ion in the K-channel at different heights (see also Figure 20d in “Materials and Methods”). The symbols mark the average height of protein residues H96 (light blue triangle 
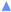
), E101 (red square 
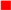
), S365 (brown triangle—right 
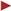
), K362 (blue cross
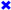
), T359 (green triangle—left 
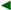
), and Y288 (magenta triangle—down 
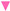
), and the position of the $H_{3}O^{+}$ ion (yellow circle
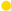
), if present. Residues with an excess proton are marked by a grey circle around the symbols representing the respective residues.

**Figure 15 ijms-24-10464-f015:**
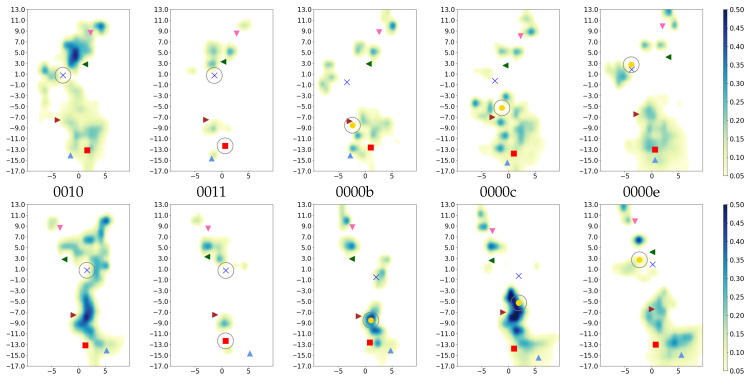
Projection of water occupancy in the K-channel of CcO with the most important K-channel protonation states (for the projections of all models, see Appendix A). The different models are labelled according to the protonation states of residues D132, E286, K362, and E101, where “0” means unprotonated and “1” means protonated (see Table 5 in “Materials and Methods”). Letters a–f refer to a $H_{3}O^{+}$ ion in the K-channel at different heights (see also Figure 20d in “Materials and Methods”). The D-channel is unprotonated in the models shown (model 00*). The lower row shows a view 90 degrees rotated from that in the upper row. For the other models, see Appendix A. The symbols mark the average height of protein residues H96 (light blue triangle 
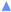
), E101 (red square 
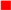
), S365 (brown triangle—right 
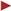
), K362 (blue cross 
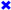
), T359 (green triangle—left 
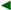
), and Y288 (magenta triangle—down 
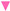
), and the position of the $H_{3}O^{+}$ ion (yellow circle 
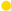
). Residues with an excess proton are marked by a grey circle around the symbols representing the respective residues.

**Figure 16 ijms-24-10464-f016:**
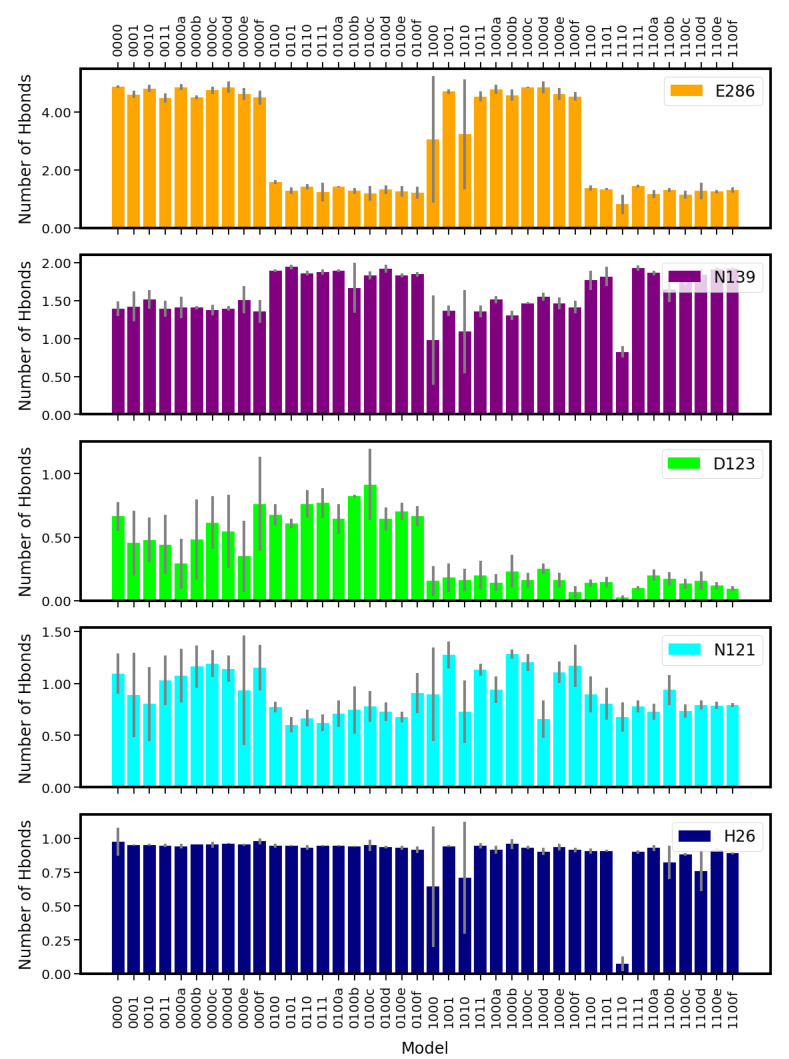
Average number of hydrogen bonds between protein residues and water in the D-channel of CcO (see also Appendix A). The different models are labelled according to the protonation states of residues D132, E286, K362, and E101, where “0” means unprotonated and “1” means protonated (see Table 5 in “Materials and Methods”). Letters a–f refer to a $H_{3}O^{+}$ ion in the K-channel at different heights (see also Figure 20d in “Materials and Methods”).

**Figure 17 ijms-24-10464-f017:**
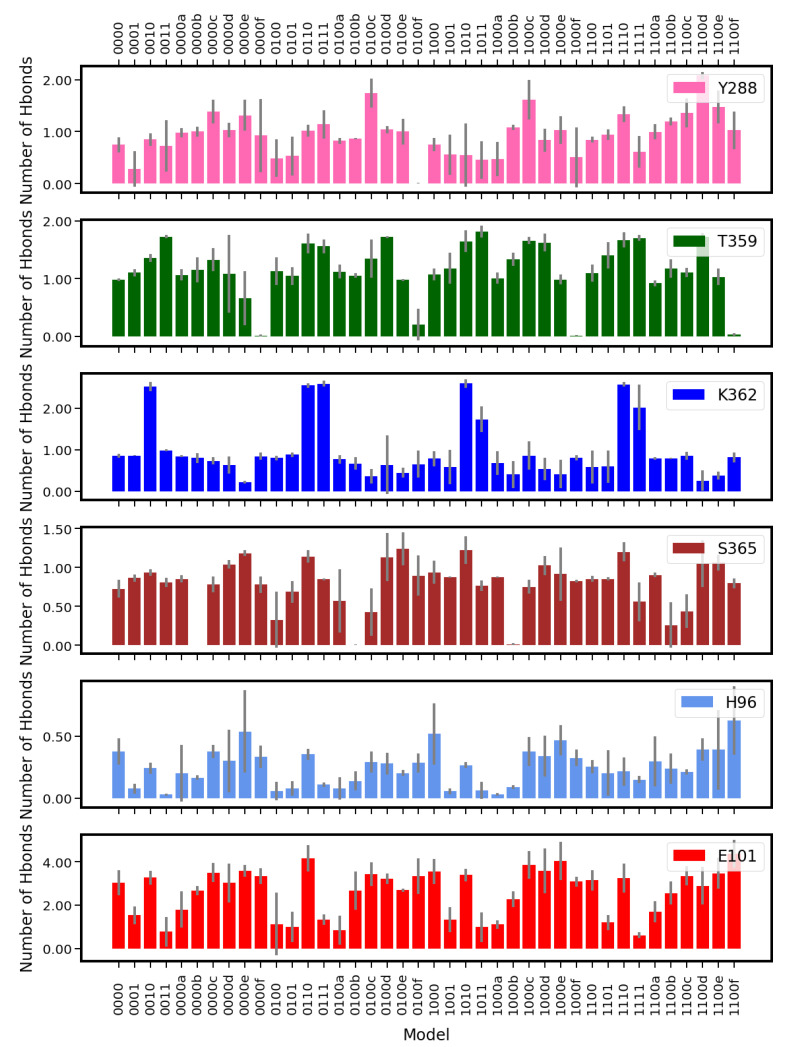
Average number of hydrogen bonds between protein residues and water in the K-channel of CcO (see also Appendix A). The different models are labelled according to the protonation states of residues D132, E286, K362, and E101, where “0” means unprotonated and “1” means protonated (see Table 5 in “Materials and Methods”). Letters a–f refer to a $H_{3}O^{+}$ ion in the K-channel at different heights (see also Figure 20d in “Materials and Methods”).

**Figure 18 ijms-24-10464-f018:**
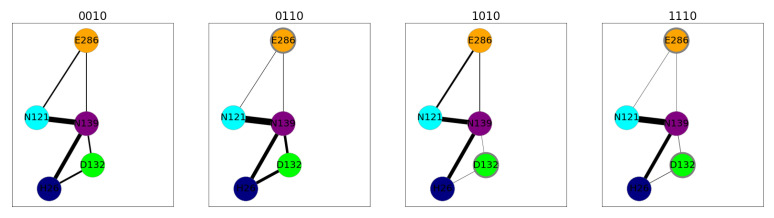
Hydrogen-bond connections between important protein residues (represented by coloured nodes: H26: dark blue; N121: cyan; D132: lime; N139: purple; E286: orange) in the D-channel of CcO for varying D-channel protonation. In the K-channel, K362 was protonated (*10 models). Residues with an excess proton are marked by a grey circle around the symbols representing the respective residues. The thickness of the lines indicates the probability of finding a hydrogen-bond connection. Only connections with at least 1% occurrence are shown for clarity. For hydrogen-bonded networks of the other models, see Appendix A. The different models are labelled according to the protonation states of residues D132, E286, K362, and E101, where “0” means unprotonated and “1” means protonated (see Table 5 in “Materials and Methods”).

**Figure 19 ijms-24-10464-f019:**
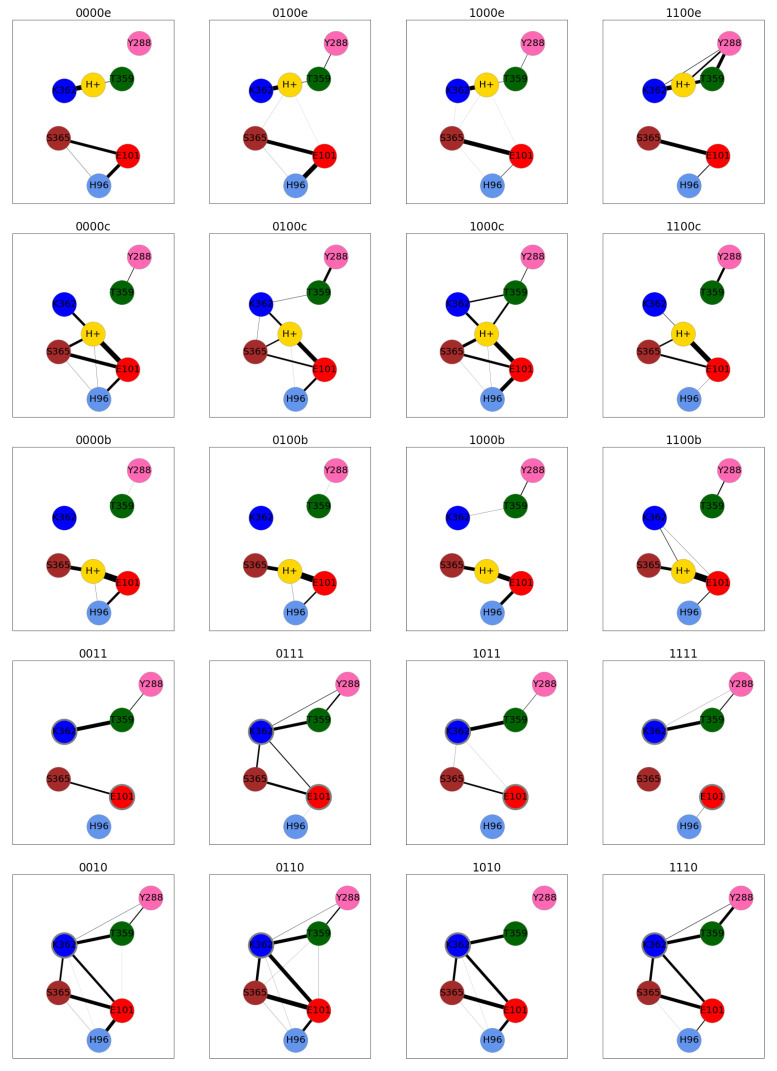
Hydrogen-bond connections between important protein residues (represented by coloured nodes: Y288: pink; T359: dark green; K362: blue; S365: brown; H96: light blue; E101: red) in the K-channel of CcO. Residues with an excess proton are marked by a grey circle around the symbols representing the respective residues. The thickness of the lines indicates the probability of finding a hydrogen-bond connection. Only connections with at least 1% occurrence are shown for clarity. For hydrogen-bonded networks of the other models, see Appendix A. The different models are labelled according to the protonation states of residues D132, E286, K362, and E101, where “0” means unprotonated and “1” means protonated (see Table 5 in “Materials and Methods”). Letters a–f refer to a $H_{3}O^{+}$ ion in the K-channel at different heights (see also Figure 20d in “Materials and Methods”).

**Table 1 ijms-24-10464-t001:** Average number of water molecules in the D-channel polyhedron (The volume of the polyhedraon is reported in Appendix A). The different models are labelled according to the protonation states of residues D132, E286, K362, and E101, where “0” means unprotonated and “1” means protonated (see Table 5 in “Materials and Methods”). Letters a–f refer to a $H_{3}O^{+}$ ion in the K-channel at different heights (see also Figure 20d in “Materials and Methods”).

00*		01*		10*		11*	
0000f	18.6 ± 0.8	0100f	13.4 ± 2.1	1000f	18.0 ± 1.6	1100f	14.3 ± 0.6
0000e	18.5 ± 0.8	0100e	13.7 ± 0.8	1000e	18.7 ± 1.8	1100e	14.3 ± 0.7
0000d	19.0 ± 0.3	0100d	13.9 ± 1.4	1000d	17.9 ± 0.5	1100d	15.3 ± 0.9
0000c	19.6 ± 1.1	0100c	14.5 ± 0.9	1000c	20.4 ± 0.5	1100c	14.6 ± 0.9
0000b	18.0 ± 1.2	0100b	13.5 ± 0.3	1000b	19.0 ± 1.0	1100b	14.8 ± 1.0
0000a	18.4 ± 0.5	0100a	14.7 ± 0.3	1000a	18.6 ± 0.4	1100a	13.9 ± 0.5
0011	17.5 ± 0.2	0111	13.8 ± 1.0	1011	17.7 ± 0.4	1111	15.1 ± 0.6
0010	18.4 ± 0.7	0110	14.4 ± 0.4	1010	18.8 ± 1.9	1110	13.9 ± 0.3
0001	17.2 ± 0.6	0101	13.9 ± 0.7	1001	19.2 ± 0.7	1101	15.1 ± 0.4
0000	19.6 ± 0.9	0100	15.6 ± 0.5	1000	18.4 ± 1.9	1100	15.2 ± 0.3

**Table 2 ijms-24-10464-t002:** Average number of water molecules in the K-channel polyhedron. (The volume of the polyhedraon is reported in Appendix A.) The different models are labelled according to the protonation states of residues D132, E286, K362, and E101, where “0” means unprotonated and “1” means protonated (see Table 5 in “Materials and Methods”). Letters a–f refer to a $H_{3}O^{+}$ ion in the K-channel at different heights (see also Figure 20d in “Materials and Methods”).

00*		01*		10*		11*	
0000f	9.1 ± 0.6	0100f	6.6 ± 1.3	1000f	7.1 ± 0.6	1100f	8.5 ± 1.2
0000e	7.1 ± 0.3	0100e	10.0 ± 2.6	1000e	9.0 ± 1.1	1100e	9.5 ± 0.5
0000d	12.4 ± 3.3	0100d	17.2 ± 1.3	1000d	17.4 ± 4.5	1100d	18.9 ± 0.8
0000c	11.5 ± 0.9	0100c	11.8 ± 1.6	1000c	14.9 ± 1.6	1100c	9.9 ± 0.4
0000b	6.7 ± 0.1	0100b	5.4 ± 1.1	1000b	5.9 ± 0.6	1100b	6.1 ± 0.5
0000a	7.2 ± 0.9	0100a	7.3 ± 1.1	1000a	6.1 ± 1.1	1100a	5.7 ± 1.0
0011	5.6 ± 0.7	0111	12.1 ± 0.5	1011	8.1 ± 1.0	1111	8.1 ± 0.9
0010	18.5 ± 0.7	0110	19.4 ± 0.9	1010	15.2 ± 2.9	1110	16.1 ± 3.8
0001	6.2 ± 0.6	0101	5.5 ± 4.1	1001	5.3 ± 1.1	1101	7.7 ± 2.3
0000	6.5 ± 1.1	0100	2.7 ± 3.4	1000	7.4 ± 1.4	1100	6.2 ± 0.7

**Table 3 ijms-24-10464-t003:** Average number of hydrogen bonds between the side chains of N121 and N139. The different models are labelled according to the protonation states of residues D132, E286, K362, and E101, where “0” means unprotonated and “1” means protonated (see Table 5 in “Materials and Methods”). Letters a–f refer to a $H_{3}O^{+}$ ion in the K-channel at different heights (see also Figure 20d in “Materials and Methods”).

00*		01*		10*		11*	
0000f	0.4 ± 0.4	0100f	0.9 ± 0.2	1000f	0.5 ± 0.3	1100f	0.9 ± 0.0
0000e	0.7 ± 0.6	0100e	1.1 ± 0.0	1000e	0.4 ± 0.2	1100e	0.9 ± 0.0
0000d	0.5 ± 0.2	0100d	1.0 ± 0.1	1000d	1.0 ± 0.3	1100d	0.9 ± 0.0
0000c	0.3 ± 0.2	0100c	0.9 ± 0.2	1000c	0.4 ± 0.1	1100c	1.1 ± 0.1
0000b	0.4 ± 0.2	0100b	1.0 ± 0.3	1000b	0.1 ± 0.0	1100b	0.8 ± 0.2
0000a	0.5 ± 0.4	0100a	1.1 ± 0.1	1000a	0.8 ± 0.2	1100a	1.0 ± 0.1
0011	0.5 ± 0.3	0111	1.1 ± 0.1	1011	0.5 ± 0.2	1111	0.9 ± 0.1
0010	0.9 ± 0.5	0110	1.0 ± 0.1	1010	0.7 ± 0.4	1110	0.9 ± 0.2
0001	0.8 ± 0.6	0101	1.1 ± 0.1	1001	0.3 ± 0.1	1101	1.0 ± 0.2
0000	0.4 ± 0.3	0100	1.0 ± 0.0	1000	0.3 ± 0.1	1100	0.9 ± 0.2

**Table 4 ijms-24-10464-t004:** Average number of hydrogen bonds between the $H_{3}O^{+}$ ion and water molecules in the K-channel of CcO. The different models are labelled according to the protonation states of residues D132, E286, K362, and E101, where “0” means unprotonated and “1” means protonated (see Table 5 in “Materials and Methods”). Letters a–f refer to a $H_{3}O^{+}$ ion in the K-channel at different heights (see also Figure 20d in “Materials and Methods”).

z	0000z	0100z	1000z	1100z
f	1.4 ± 0.6	1.3 ± 0.4	1.5 ± 0.5	1.3 ± 0.3
e	1.0 ± 0.1	1.1 ± 0.2	1.3 ± 0.3	1.4 ± 0.2
c	2.5 ± 0.1	2.6 ± 0.2	2.6 ± 0.1	2.3 ± 0.2
d	2.2 ± 0.2	2.2 ± 0.4	2.0 ± 0.1	1.8 ± 0.1
b	1.2 ± 0.2	1.1 ± 0.4	0.8 ± 0.4	1.3 ± 0.5
a	1.5 ± 0.2	1.0 ± 0.7	1.2 ± 0.2	1.4 ± 0.2

## Data Availability

The data presented in this study are available upon request from the corresponding author.

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
