# Peer review of "Protonation-State Dependence of Hydration and Interactions in the Two Proton-Conducting Channels of Cytochrome c Oxidase"

_ijms, 2023, doi:10.3390/ijms241310464_

Round 1

Reviewer 1 Report

The paper covers molecular dynamics simulations of proton pumping in cytochrome oxidase and probes the possible connection between the K and D channel amino acids in the pumping.

Major issues

11)     I respect the sheer amount of data that each of these simulations generates.  However, I found it extremely difficult to look at the data itself in the figures and tables and correlate to the statements made in the text of the results.  For example, lines 282-283 it took quite a while to discern the model numbering.  I would suggest adding a short paragraph at the start of the results that gives the reader more orientation as to how the numbering system works.  It is in the table in the materials and methods but is never explained in the results and discussion.

22)     Continuing with point 1- I would suggest including the models used for each statement about amino acid interactions or hydration levels. 

33)     The paragraph that encompasses lines 383-401 to me was almost incomprehensible.  I could not follow the models used to make almost any of the statements and then correlating them to values in Tables 2 and 3.   For example, it says on line 385 that there is high hydration in models *11, yet in table 3, the *11 models have the lower numbers. The higher hydration values were in models *10- are the model numbers wrong or is the statement wrong?

44)     In line 280-288, there are statements about the difference in distances between the residues.  However, when I look at the data in figure 4, there are some in the 00* models that have sub-populations with short distances.  Are those just ignored to make a statement about the major population?  Or should they also be considered?

55)     Some figures are in the paper but not referenced in the text anywhere.  Figure 5, Figure 8, Figure 9, Figure 10, Figure 13, Figure 14, Figure 15, Figure 16, Figure 19.  If these are important enough to be figures in the paper, they should be reference and explained in the text of the paper.  If not, move them to supplementary material. I had a hard time figuring out what they were and why they were there since there was not text to support them.

Minor issues

66)     There are several one sentence paragraphs.  These paragraphs should have more than one sentence or be incorporated into the paragraph before or after.  One example- lines 58-60, that sentence could be incorporated into the following paragraph. (Also lines 298-300, 436-439 are two others I marked but there may be more)

77)     Lines 39-42- I found this short paragraph hard to read. I suggest taking out the “in contrast” in the middle of line 41 to help with clarity.

88)     Line 56 proton should be plural “…protons located in the lower….”

99)     Line 62 change sentence to read “….channel to contain sufficient water molecules such that…” to make more clear.

110)  Line 298 first word should be “The” not “These”

111)  Line 304 end of line result should be results- to read “The results are when K362”

112)  Line 322 It has Table ??- that should refer to an actual table not ??

113)  Table 3- the 11* should be above the lines with the models not the hydration level column

114)  Line 534- the start of the line should have “the” before K-channel so it reads “in the K-channel”

115) Line 557 it says “see Figures 16 and 16”- is that supposed to refer to another different numbered figure or supplementary figure 16?

Please make the changes in the minor issues section

Author Response

Please see attached file "response1.pdf" .

Reviewer 2 Report

The main question addressed by the research is the possible interconnection of the protonation state in one of the two possible channels of proton transfer in Cytochrome c Oxidase (CcO) on the properties of the other channel. The topic of this study has its own history and prior to this research the question about the operations of D and K channels remained unanswered. For example, the authors mentioned that pH gradient studies across the membrane showed that the protonation of the key residues in the two channels is not independent of each other, though, the simultaneous protonation of Y288 and E286 has been found to be unlikely and this points at the fact that the D-channel can therefore take place only before arrival of the "chemical" proton at the Y288 in the K-channel. This study sheds some light on this issue, and therefore can be considered as original and filling a specific gap in the field of biochemistry of aerobic respiration that involves the mechanism of molecular oxygen reduction to water, being a very exergonic reaction, which results in conserving of a substantial part of the free energy as an electrochemical gradient over the mitochondrial or bacterial membrane. The topic of the study can also be said to be relevant to the Special Issue “Ion Pumps: Molecular Mechanisms, Structure, Physiology” of the “Molecular Biology” section of IJMS. The authors carried out a gigantic work to scrutiny the issue, they have resorted to the molecular dynamic simulations, performed the conformational analysis of the side chain protein residues, carried out the computation of electrostatic interactions between different pairs of residues, identified the hydration of the K and D channels using the topological analysis, and so on. I won’t list all the aspects of their work as these are so many. As a result, the authors established that no significant effect of the protonation state in one channel on the properties of the other channel could be observed, and the regulation of proton transport through the two channels in the correct order must therefore be achieved via other media such as the electron transfer to the bi-nuclear center. This work represents a valuable contribution to the cellular biology and serves as an example for future publications in the field of biochemistry. The text is clearly written and English is very good. No issues were detected in the literature list. Only one misprint was found in L. 322: “Table ??.” Overall, I’ve received a good impression from this work, I believe that it would be interesting to a broad audience interested in cellular biochemistry and I recommend this work for publication after minor spell checking.   

Author Response

Please see attached file "response2.pdf" .
